# Efficient Multi-Agent Cooperation Learning through Teammate Lookahead

**Feng Chen**[1,2], **Xinwei Chen**[2], **Rongjun Qin**[1,2], **Cong Guan**[1], **Lei Yuan**[1,2],
**Zongzhang Zhang**[1],* **Yang Yu**[1,2]

[1] *National Key Laboratory for Novel Software Technology, Nanjing University*
  *School of Artificial Intelligence, Nanjing University*
[2] *Polixir Technologies*

*chenf@lamda.nju.edu.cn, xinwei.chen@polixir.ai,*
*{qinrj, guanc, yuanl}@lamda.nju.edu.cn, {zzzhang, yuy}@nju.edu.cn*

**Reviewed on OpenReview:** *https://openreview.net/forum?id=CeNNIQ8GJf*

## Abstract

Cooperative Multi-Agent Reinforcement Learning (MARL) is a rapidly growing research field that has achieved outstanding results across a variety of challenging cooperation tasks. However, existing MARL algorithms typically overlook the concurrent updates of teammate agents. An agent always learns from the data that it cooperates with one set of (current) teammates, but then practices with another set of (updated) teammates. This phenomenon, termed as "teammate delay", leads to a discrepancy between the agent's learning objective and the actual evaluation scenario, which can degrade learning stability and efficiency. In this paper, we tackle this challenge by introducing a lookahead strategy that enables agents to learn to cooperate with predicted future teammates, allowing the explicit awareness of concurrent teammate updates. This lookahead strategy is designed to seamlessly integrate with existing policy-gradient-based MARL methods, enhancing their performance without significant modifications to their underlying structures. The extensive experiments demonstrate the effectiveness of this approach, showing that the lookahead strategy enhances cooperation learning efficiency and achieves competitive performance compared to state-of-the-art MARL algorithms.

## 1 Introduction

Cooperative Multi-Agent Reinforcement Learning (MARL) techniques focus on replicating the collaborative intelligence observed in human teams (Oroojlooy & Hajinezhad, 2023), and advancements in recent years have showcased its remarkable potential in various application domains, including robotics (Wang et al., 2022), games (Berner et al., 2019), and social networks (Leibo et al., 2017). Among them, multi-agent policy gradient methods stand out with the capability to handle continuous control tasks and with potential to solve intricate cooperative problems (de Witt et al., 2020a; Yu et al., 2022a). Despite the ongoing progresses in this category of methods, we point out that they typically suffer from a "teammate delay" issue. Specifically, this issue occurs when an agent learns from the data that it cooperates with current teammates, but then practices with updated teammates due to the concurrent teammate updates. It is worth noting that the "teammate delay" phenomenon is related to, yet distinct from, the general non-stationarity issue (Yuan et al., 2023) in multi-agent systems. Whereas non-stationarity broadly refers to learning challenges arising from dynamic policy changes across all agents (including both teammates and opponents), the identified "teammate delay" specifically characterizes learning inefficiency in cooperative settings caused by agents neglecting teammate

---

*Corresponding Author

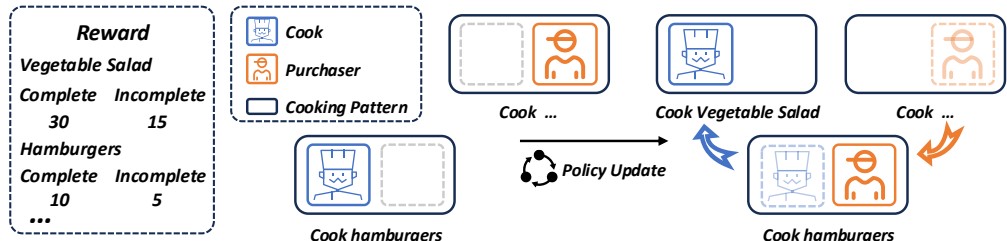

Figure 1: One simple example to show "teammate delay" issue, where *Purchaser* is expected to buy corresponding ingredients for *Cook* to cook.

policy updates. This problem is particularly pronounced in on-policy MARL algorithms, where agents optimize their policies based on outdated teammate policies.

For more intuitive illustration, one concrete example is shown in Figure 1. In this example, *Cook* initially wants to cook hamburgers, and *Purchaser* adjusts its policy to buy corresponding ingredients after one round of policy update. However, at this moment, *Cook* also improves its policy to cook vegetable salad. Thereby, their updated policies fail to cooperate well. This simple example reveals that updating the agents to cooperate with current teammates would lead to a training-test mismatch because the teammates update their policies as well. This gap between policy training and evaluation in each round of update can lead to severe learning inefficiency.

Although not explicitly pointing out this teammate delay issue, there exist works aiming to resolve similar problems arising from concurrent updates of teammate policies. Opponent modeling methods seek to alleviate the non-stationarity in multi-agent scenarios through explicitly modeling the teammate policies (Yuan et al., 2023). They either introduce an auxiliary task of predicting teammate behaviors (Hernandez-Leal et al., 2019) or learn teammate representations as extra policy conditions (Papoudakis & Albrecht, 2020; Cao et al., 2023). Despite their effectiveness in many problem scenarios, these methods necessitate extra teammate modeling efforts and lack theoretical analysis support. On the other hand, recent works, LOLA (Foerster et al., 2018a) and COLA (Willi et al., 2022), explicitly acknowledge the learning behavior of other agents and propose learning rules with opponent-learning awareness. However, these works are limited to two-player simple problems and face challenges to extend to practical cooperative scenarios. The most promising approach is one recent progress of multi-agent policy gradient method, HAPPO (Kuba et al., 2021). This method proposes a sequential policy update scheme with theoretical guarantees for joint policy improvement. However, its actual implementation involves an approximation utilizing *importance sampling*, potentially influencing the actual performance due to large variance in policy gradients.

Despite all these previous efforts, how the teammate delay issue influences the cooperative policy learning and how to better mitigate its negative impact are still open questions. To answer these two questions, in this paper, we both provide a formal analysis about the impact of this issue on the policy update, which motivates us to predict future teammate policies, and propose a model-based MARL algorithm where we approximate the future teammates via conducting policy updates within the environment model. In summary, our main contributions are:

- We offer a rigorous formal analysis on policy-gradient MARL algorithms by investigating the regret of the updated policy, which unveils the impact of "teammate delay" issue on cooperative policy learning.

- Furthermore, we introduce a practical model-based MARL algorithm explicitly designed to address the challenges posed by the "teammate delay" issue. By leveraging insights from our formal analysis, our algorithm aims to enhance cooperative policy learning.

- To validate the effectiveness of our proposed approach, we conduct empirical studies on various benchmarks. These benchmarks include complex problems with continuous action spaces, as well as challenging multi-agent cooperative tasks. The empirical results validate the effectiveness of our

method, showcasing its ability to match or exceed the performance of existing approaches across diverse scenarios with performance gains.

## 2 Preliminaries

In this section, we will introduce some basic preliminaries concerning our problem formulation and analysis. We will firstly introduce these concepts in single-agent setting for brevity, and they can be simply extended to multi-agent setting.

In **single-agent setting**, the sequential decision-making problem can be modeled as a Markov decision process (MDP). It can be defined as a tuple $(\mathcal{S}, \mathcal{A}, \mathcal{P}, \mathcal{R}, \gamma, \rho_0)$, where $\mathcal{S}$ is the state space, $\mathcal{A}$ is the action space, $\mathcal{P}(\cdot|s, a)$ means the transition function, $\mathcal{R}(s, a)$ is the reward function, $\gamma \in (0, 1)$ is the discounted factor, while $\rho_0$ is the initial state distribution. The agent policy is defined as $\pi(a|s)$ and the goal of Reinforcement Learning (RL) is to maximize the expected discounted return:

$$\eta(\pi) = \mathbb{E}_{s_0 \sim \rho_0, a_t \sim \pi(\cdot|s_t), s_t \sim \mathcal{P}(\cdot|s_{t-1}, a_{t-1})} \left[ \sum_{t=0}^{\infty} \gamma^t \mathcal{R}(s_t, a_t) \right]. \tag{1}$$

Moreover, for a given agent policy $\pi$, we can define the stationary state distribution at timestep $t$ as $d_t^\pi(s) = P_\pi(s_t = s)$. Then, the discounted state visitation distribution can be defined as $d_\pi(s) = (1-\gamma)\sum_{t=0}^{\infty} \gamma^t d_t^\pi(s)$, and the discounted state-action visitation distribution can be defined as $d^\pi(s, a) = d_\pi(s)\pi(a|s)$. Based on these definitions, the expected discounted return of policy $\pi$ can be rewritten as:

$$\eta(\pi) = \frac{1}{1-\gamma} \sum_{s \in \mathcal{S}} d_\pi(s) \sum_{a \in \mathcal{A}} \pi(a|s)\mathcal{R}(s, a). \tag{2}$$

For the sake of clarity in the subsequent derivations, we additionally define $\rho_\pi(s) = \sum_{t=0}^{\infty} \gamma^t d_t^\pi(s) = (\frac{1}{1-\gamma})d_\pi(s)$, which can be regarded as the unnormalized version of $d_\pi(s)$. Then the expected return further equals to $\eta(\pi) = \sum_{s \in \mathcal{S}} \rho_\pi(s) \sum_{a \in \mathcal{A}} \pi(a|s)\mathcal{R}(s, a)$. We define the agent policy that can maximize $\eta(\pi)$ as the optimal policy $\pi^*$, which means that $\eta(\pi^*) \geq \eta(\pi)$ for all Markovian agent policy $\pi$.

Moreover, for two different agent policy $\pi'$ and $\pi$, their difference in discounted return is given by the famous **performance discrepancy lemma**:

$$\eta(\pi') - \eta(\pi) = \sum_{s \in \mathcal{S}} \rho_{\pi'}(s) \left[ \sum_{a \in \mathcal{A}} \pi'(a|s)A_\pi(s, a) \right], \ A_\pi(s, a) = Q_\pi(s, a) - V_\pi(s, a), \tag{3}$$

$$Q_\pi(s, a) = \mathbb{E}_\pi \left[ \sum_{t=0}^{\infty} \gamma^t \mathcal{R}(s_t, a_t)|s_0 = s, a_0 = a \right], \ V_\pi(s) = \sum_{a \in \mathcal{A}} \pi(a|s)Q_\pi(s, a), \tag{4}$$

where $Q_\pi(s, a)$ and $V_\pi(s)$ are respectively the state-action value function and state value function.

For **multi-agent setting**, most definitions are similar. The main difference lies in the fact that there are $N$ agents in the multi-agent setting, leading to the concepts of joint action space $\boldsymbol{\mathcal{A}}$ and joint policy $\boldsymbol{\pi}$. Accordingly, the discounted return and performance discrepancy lemma (Kakade & Langford, 2002) become:

$$\eta(\boldsymbol{\pi}) = \sum_{s \in \mathcal{S}} \rho_\pi(s) \sum_{\boldsymbol{a} \in \boldsymbol{\mathcal{A}}} \boldsymbol{\pi}(\boldsymbol{a}|s)\mathcal{R}(s, \boldsymbol{a}), \tag{5}$$

$$\eta(\boldsymbol{\pi}') - \eta(\boldsymbol{\pi}) = \sum_{s \in \mathcal{S}} \rho_{\boldsymbol{\pi}'}(s) \left[ \sum_{\boldsymbol{a} \in \boldsymbol{\mathcal{A}}} \boldsymbol{\pi}'(\boldsymbol{a}|s)A_{\boldsymbol{\pi}}(s, \boldsymbol{a}) \right]. \tag{6}$$

Notably, we focus on the fully cooperative setting, where the environmental reward $\mathcal{R}(s, \boldsymbol{a}$ is shared among all agents. Additionally, whether in the single-agent or multi-agent setting, the problem formulation only considers Markovian policies, which means the agent policy makes decisions based on the state information at the current timestep. There is some slight abuse of notation, as the subscript $i$ will also denote the $i$-th agent in subsequent derivations. For example, $\pi_i$ represents the policy of the $i$-th agent. More details about the notations are listed in Appendix A.1.

### 2.1 Single-Agent Policy Gradient

In single-agent setting, the goal of reinforcement learning is to maximize the discounted return $\eta(\pi)$ of the agent policy $\pi$, which according to the performance discrepancy lemma equals to:

$$\eta(\pi) = \eta(\pi^k) + \sum_{s \in \mathcal{S}} \rho_\pi(s) \left[ \sum_{a \in \mathcal{A}} \pi(a|s) A_{\pi^k}(s, a) \right], \ A_{\pi^k}(s, a) = Q_{\pi^k}(s, a) - V_{\pi^k}(s), \tag{7}$$

where $\pi^k$ is the $k$-th round agent policy. As it is hard to sample trajectories corresponding to the state distribution $\rho_\pi(s)$, in practice, we typically use $\pi^k$ to sample trajectories instead. This implies that the actual learning objective is:

$$J(\pi) = \eta(\pi^k) + \sum_{s \in \mathcal{S}} \rho_{\pi^k}(s) \left[ \sum_{a \in \mathcal{A}} \pi(a|s) A_{\pi^k}(s, a) \right]. \tag{8}$$

In fact, $J(\pi)$ is related to $\pi^k$, but for brevity we omit it in input. The same applies to the following context. Due to the state distribution change, $\pi$ can not be updated too far away from $\pi^k$, for which traditional actor-critic algorithms, e.g., A3C (Mnih et al., 2016), conduct only a few policy gradient ascend while trust-region algorithms, such as PPO (Schulman et al., 2017), conforms to the trust-region optimization.

### 2.2 Multi-Agent Policy Gradient

When it comes to the multi-agent setting, the problem can be defined as a tuple $(N, \mathcal{S}, \mathcal{A}, \mathcal{P}, \mathcal{R}, \gamma, \rho_0)$, where $N$ is the number of agents. It should be noted that in some scenarios, the agents need to make decisions based on local observations. For the sake of simplicity and without loss of generality, we have omitted the partial observability in derivation. Moreover, we assume that the joint policy $\boldsymbol{\pi}$ can be decomposed into the product of individual policies $\boldsymbol{\pi}(\boldsymbol{a}|s) = \prod_{i=1}^{N} \pi_i(a_i|s)$, where $\pi_i$ is the individual policy for agent $i$ and the joint action $\boldsymbol{a} = [a_1, a_2, \cdots, a_N] \in \mathcal{A}$ is decomposed of individual actions $\{a_i\}_{i=1}^{N}$.

In this case, the learning objective for each agent in multi-agent policy gradient methods is typically defined as:

$$\begin{aligned} &J_i(\pi_i | \{\pi_j^k\}_{j \neq i}) \\ &= \eta(\boldsymbol{\pi}^k) + \sum_{s \in \mathcal{S}} \rho_{\boldsymbol{\pi}^k}(s) \left[ \sum_{\boldsymbol{a} \in \mathcal{A}} \pi_i(a_i|s) \prod_{j \neq i} \pi_j^k(a_j|s) A_{\boldsymbol{\pi}^k}(s, \boldsymbol{a}) \right], \ i \in \{1, 2, \cdots, N\}. \end{aligned} \tag{9}$$

More detailed derivation for this learning objective is provided in Appendix A.2. Compared with that in single-agent setting, the data distribution here is also influenced by the teammate policies. In other words, the $i$-th agent updates its policy associated with the current teammates $\{\pi_j^k\}_{j \neq i}$ in this learning objective. Overall, the learning objective for the joint policy can be formalized as:

$$J(\boldsymbol{\pi}) = \eta(\boldsymbol{\pi}^k) + \sum_{s \in \mathcal{S}} \rho_{\boldsymbol{\pi}^k}(s) \left[ \frac{1}{N} \sum_{i=1}^{N} \sum_{\boldsymbol{a} \in \mathcal{A}} \pi_i(a_i|s) \prod_{j \neq i} \pi_j^k(a_j|s) A_{\boldsymbol{\pi}^k}(s, \boldsymbol{a}) \right]. \tag{10}$$

## 3 Method

In this work, we identify the teammate delay phenomenon in the common practice of multi-agent policy gradient methods. The direct negative impact of this issue can be analyzed and how to solve this issue deserves further study. In this section, we firstly discuss how the teammate delay issue can cause a negative impact on the cooperative policy learning through analyzing the regret of the updated joint policy at the next round. Motivated by this analysis, we then propose a practical algorithm that exploits the future teammate information to facilitate the cooperation learning.

### 3.1 Analysis Motivates Predicting Future Teammates

From Equation (10), we know that in typical multi-agent policy gradient methods, the learning objective of the agent policy involves computing an expectation with respect to the current teammate policies $\{\pi_j^k\}_{j \neq i}$. Consequently, the current policy distribution of the teammates will have an impact on the policy update. In order to provide further analysis on this impact, we replace the teammate policies with a general notation $\{\mu_j\}_{j \neq i}$, which means that the trajectories are sampled associated with a sampling policy $\boldsymbol{\mu}$. In this way, the learning objective is transformed into:

$$J(\boldsymbol{\pi}, \boldsymbol{\mu}) = \eta(\boldsymbol{\pi}^k) + \sum_{s \in \mathcal{S}} \rho_{\boldsymbol{\mu}}(s) \left[ \frac{1}{N} \sum_{i=1}^N \sum_{\boldsymbol{a} \in \mathcal{A}} \pi_i(a_i|s) \prod_{j \neq i} \mu_j(a_j|s) A_{\boldsymbol{\pi}^k}(s, \boldsymbol{a}) \right], \tag{11}$$

where $\boldsymbol{\pi}^k$ still denotes the joint policy at the $k$-th round. In existing multi-agent policy-gradient methods, the sampling policy $\boldsymbol{\mu}$ is typically selected to be $\boldsymbol{\pi}^k$, which means that we expect $\pi_i$ to collaborate well with the $k$-th round teammate policies through maximizing $\sum_{s \in \mathcal{S}} \rho_{\boldsymbol{\pi}^k}(s) \sum_{\boldsymbol{a} \in \mathcal{A}} \pi_i(a_i|s) \prod_{j \neq i} \pi_j^k(a_j|s) A_{\boldsymbol{\pi}^k}(s, \boldsymbol{a})$.

We wonder what would happen when we adjust $\boldsymbol{\mu}$ from $\boldsymbol{\pi}^k$ to other distributions. To answer this question, we firstly propose the following lemma that estimates the upper bound of discrepancy between the learning objective $J(\boldsymbol{\pi}, \boldsymbol{\mu})$ and the actual policy return $\eta(\boldsymbol{\pi})$.

**Lemma 1** *Assume that we update the joint policy $\boldsymbol{\pi}^k$ to $\boldsymbol{\pi}^{k+1}$ with sampling policy $\boldsymbol{\mu}$. Given the measurement of distance between sampling policy $\boldsymbol{\mu}$ and the updated policy $\boldsymbol{\pi}^{k+1}$ as $\alpha_i = \max_s D_{\mathrm{TV}} \left( \pi_i^{k+1}(\cdot|s) \| \mu_i(\cdot|s) \right)$ [1], we have:*

$$|J(\boldsymbol{\pi}^{k+1}, \boldsymbol{\mu}) - \eta(\boldsymbol{\pi}^{k+1})| \leq \frac{4\epsilon\gamma}{(1-\gamma)^2} \left( \sum_{i=1}^N \alpha_i \right)^2 + \frac{2\epsilon(N-1)}{N} \sum_{i=1}^N \alpha_i, \tag{12}$$

*where $\gamma$ is the discount factor and $\epsilon = \max_{s, \boldsymbol{a}} |A_{\boldsymbol{\pi}^k}(s, \boldsymbol{a})|$.*

For proof see Appendix A.2. The estimated upper bound of the discrepancy between $J(\boldsymbol{\pi}^{k+1}, \boldsymbol{\mu})$ and $\eta(\boldsymbol{\pi}^{k+1})$ in Lemma 1 can aid us in analyzing the regret of $\boldsymbol{\pi}^{k+1}$, leading to the following theorem:

**Corollary 1** *Suppose that we update joint policy $\boldsymbol{\pi}^k$ to $\boldsymbol{\pi}^{k+1}$ with sampling policy $\boldsymbol{\mu}$, then the regret of the updated joint policy $\boldsymbol{\pi}^{k+1}$ has the following upper bound:*

$$\eta(\boldsymbol{\pi}^*) - \eta(\boldsymbol{\pi}^{k+1}) \leq \eta(\boldsymbol{\pi}^*) - J(\boldsymbol{\pi}^{k+1}, \boldsymbol{\mu}) + \underbrace{\frac{4\epsilon\gamma}{(1-\gamma)^2} \left( \sum_{i=1}^N \alpha_i \right)^2 + \frac{2\epsilon(N-1)}{N} \sum_{i=1}^N \alpha_i}_{(c)}. \tag{13}$$

For proof see Appendix A.2. The right hand side of Inequality (13) sheds light on the elements that can influence the cooperative policy learning. Totally, we expect to minimize the regret of the updated policy via minimizing the overall upper bound expression. Other than the first term $\eta(\boldsymbol{\pi}^*)$ that is a constant value, the upper bound is composed of $-J(\boldsymbol{\pi}^{k+1}, \boldsymbol{\mu})$ and one extra term $(c)$. In fact, the second term $-J(\boldsymbol{\pi}^{k+1}, \boldsymbol{\mu})$ is exactly the loss function that the algorithm aims to minimize at each update round, which typically serves as a surrogate function for the regret $\eta(\boldsymbol{\pi}^*) - \eta(\boldsymbol{\pi}^{k+1})$. However, Inequality (13) reveals that the regret can not be bounded by $-J(\boldsymbol{\pi}^{k+1}, \boldsymbol{\mu})$ alone, and an extra term $(c)$ relatively captures the gap between this surrogate function and the actual regret.

This extra term $(c)$, a function of the sampling policy $\boldsymbol{\mu}$ and the updated policy $\boldsymbol{\pi}^{k+1}$, can not be optimized by the previous learning algorithms, but it can have an impact on the cooperation learning. When given a large term $(c)$, the surrogate function would be far from the actual regret, which can result in low learning efficiency. In fact, it is easy to observe that term $(c)$ will be reduced to zero when $\boldsymbol{\mu}$ is equivalent to $\boldsymbol{\pi}^{k+1}$.

---

[1] The TV distance measures the distance between two distributions via calculating $D_{\mathrm{TV}}(P, Q) = \frac{1}{2} \sum_x |P(x) - Q(x)|$ (Cover, 1999).

---

**Algorithm 1** Multi-Agent Policy Gradient Learning with Lookahead

---

**Input**: The number of agent $N$, max iteration number $K$, trajectory batch size $M$
**Output**: Obtained multi-agent cooperation policy

1: Initialize replay buffer $\mathcal{B}$;
2: Initialize a joint policy $\boldsymbol{\pi} = \{\pi_i\}_{i=1}^N$ randomly;
3: **for** iteration $k = 1$ to $K$ **do**
4:     Sample a batch of transitions from $\mathcal{B}$ and update the environment model by minimizing loss $\mathcal{L}_{\text{model}}$;
5:     Sample a batch of trajectories $\{\tilde{\tau}\}_M$ in the environment model with sampling policy $\boldsymbol{\pi}^k$, and obtain $\tilde{\boldsymbol{\pi}}^{k+1} = \psi(\boldsymbol{\pi}^k, \boldsymbol{\pi}^k)$ using the training trajectories $\{\tilde{\tau}\}_M$;
6:     Sample a batch of trajectories $\{\tau\}_M$ in the real environment with sampling policy $\tilde{\boldsymbol{\pi}}^{k+1}$, and obtain $\boldsymbol{\pi}^{k+1} = \psi(\tilde{\boldsymbol{\pi}}^{k+1}, \boldsymbol{\pi}^k)$ using the training trajectories $\{\tau\}_M$;
7:     Add trajectories $\tau$ to the buffer $\mathcal{B}$;
8: **end for**

---

That is, the extra term $(c)$ would disappear if we trained the agents with the information of future teammates. This outcome motivates us to replace the sampling policy $\boldsymbol{\mu}$ with an approximation of the future teammates, thus to reduce the regret. A more comprehensive analysis on how approximating future teammates reduces the regret upper bound is provided in Appendix A.3. In Section 3.2 and Section 3.3, we will show how we approximate the future teammates.

## 3.2 Future Teammate Approximation

Based on the above analysis, we are motivated to replace the sampling policy $\boldsymbol{\mu}$ with the future teammate policy $\boldsymbol{\pi}^{k+1}$ in each round of policy update. However, achieving this goal is not easy in practice, because in each round of policy update, the updated policy $\boldsymbol{\pi}^{k+1}$ is affected by the sampling policy $\boldsymbol{\mu}$, that is, $\boldsymbol{\pi}^{k+1}$ and $\boldsymbol{\mu}$ are coupled. Thus, to serve this goal, we propose that the future teammate policy can be obtained by solving a bi-level optimization problem below:

**Theorem 1** *Suppose the sampling policy $\boldsymbol{\mu}^*$ can derive the same updated policy, which means that $\boldsymbol{\mu}^* = \arg\max_{\boldsymbol{\pi}} J(\boldsymbol{\pi}, \boldsymbol{\mu}^*)$. If it exists, it will be the solution of the following bi-level optimization problem:*

$$\arg\min_{\boldsymbol{\mu}} D_{\text{KL}}(\boldsymbol{\mu} \| \boldsymbol{\pi}^{k+1}), \;\; s.t. \;\; \boldsymbol{\pi}^{k+1} = \arg\max_{\boldsymbol{\pi}} J(\boldsymbol{\pi}, \boldsymbol{\mu}). \tag{14}$$

Its proof can be found in Appendix A.2. This theorem inspires us that we can obtain the expected $\boldsymbol{\mu}^*$ by solving the consistent bi-level optimization problem. The solution of this problem to some extent contains the information of future teammate policy. However, this problem typically follows a form of Stackelberg Game (Friedman, 1971), and is not easy to solve.

In this case, we propose to perform one-step approximation of this optimization problem, which means that with $\boldsymbol{\mu}$ initialized as $\boldsymbol{\pi}^k$, we firstly solve the inner-loop optimization with $\boldsymbol{\pi}^{k+1} = \arg\max_{\boldsymbol{\pi}} J(\boldsymbol{\pi}, \boldsymbol{\mu})$ and then we assign the obtained $\boldsymbol{\pi}^{k+1}$ to the sampling policy $\boldsymbol{\mu}$, thus obtaining the approximation of the solution $\boldsymbol{\mu}^*$. This one-step approximation is commonly utilized for stackelberg-game-like problems, and it achieves a trade-off between the solution accuracy and the computation cost. In brief, for feasible future teammate approximation, we perform an additional round of optimization before each algorithm iteration, using the previous round's policy $\boldsymbol{\pi}^k$ as the sampling policy, and the obtained policy $\tilde{\boldsymbol{\pi}}^{k+1}$ serves as the approximation of future teammate policy.

## 3.3 Practical Algorithm Implementation

**Model-based Approximation**  The above analysis motivates us to conduct extra training to estimate the future teammate policy. However, a straight-forward implementation is not practical because it wastes near half of the online samples for estimating the future teammate policy, and those samples are not utilized for the actual policy training, which as a result will lead to very low sample efficiency of the algorithm. To

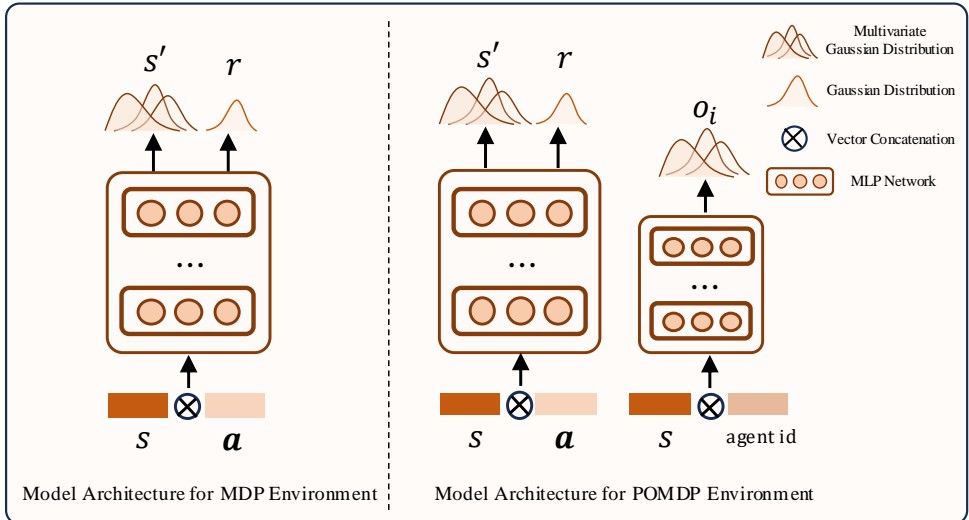

Figure 2: Network architectures for environment model in practical implementation.

avoid this issue, we propose to learn an environment model, and put the teammate policy estimation process within it, thus avoiding the waste of a large number of online samples.

In practice, we build the environment model utilizing multiple Multi-Layer Perceptrons (MLPs), as shown in Figure 2. It takes the environment state $s$, joint action $\boldsymbol{a}$ as input, and predicts the distributions of next state $s'$ and reward signal $r$. Specifically, the next state is modeled as a multivariate Gaussian distribution $s' \sim \mathcal{N}(\mu_s(s, \boldsymbol{a}|\theta), \Sigma_s(s, \boldsymbol{a}|\theta))$, and the reward is modeled as a univariate Gaussian $r \sim \mathcal{N}(\mu_r(s, \boldsymbol{a}|\theta), \sigma_r(s, \boldsymbol{a}|\theta)^2)$. To train the environment model, we construct a replay buffer $\mathcal{B}$ and store the online samples into it throughout the training process. At each iteration, we will sample data batches from buffer $\mathcal{B}$ to minimize the following negative log-likelihood of the true data:

$$\mathcal{L}_{\text{model}}(\theta) = \mathcal{L}_{\text{trans}}(\theta) + \mathcal{L}_{\text{reward}}(\theta) \tag{15}$$

$$\mathcal{L}_{\text{trans}}(\theta) = \mathbb{E}_{(s,\boldsymbol{a},s',r)\sim\mathcal{B}} \left[ -\log p(s'|\mu_s, \Sigma_s) \right] = \mathbb{E}_{(s,\boldsymbol{a},s',r)\sim\mathcal{B}} \left[ \frac{1}{2} \left( (s'-\mu_s)^\top \Sigma_s^{-1}(s'-\mu_s) + \log|\Sigma_s| \right) \right], \tag{16}$$

$$\mathcal{L}_{\text{reward}}(\theta) = \mathbb{E}_{(s,\boldsymbol{a},s',r)\sim\mathcal{B}} \left[ -\log p(r|\mu_r, \sigma_r^2) \right] = \mathbb{E}_{(s,\boldsymbol{a},s',r)\sim\mathcal{B}} \left[ \frac{1}{2} \left( \frac{(r-\mu_r)^2}{\sigma_r^2} + \log \sigma_r^2 \right) \right], \tag{17}$$

where $\mu_s, \Sigma_s, \mu_r, \sigma_r$ are short for $\mu_s(s, \boldsymbol{a}|\theta), \Sigma_s(s, \boldsymbol{a}|\theta), \mu_r(s, \boldsymbol{a}|\theta), \sigma_r(s, \boldsymbol{a}|\theta)$ that are the predictions of environment model $f_\theta$.

Moreover, considering that many multi-agent scenarios are partially observable, where agents can only observe a portion of the environment's state, for such POMDP environments, we additionally introduce one projection network $h_\phi$ that predicts individual observations $o_i$ from the global state $s$. Thus, for these POMDP cases, the training loss includes one additional term of observation prediction:

$$\mathcal{L}_{\text{model}}(\theta, \phi) = \mathcal{L}_{\text{trans}}(\theta) + \mathcal{L}_{\text{reward}}(\theta) + \mathcal{L}_{\text{projection}}(\phi), \tag{18}$$

$$\mathcal{L}_{\text{projection}}(\phi) = \mathbb{E}_{(s,\{o_i\}_{i=1}^N)\sim\mathcal{B}} \left[ \sum_{i=1}^N \left( -\log p(o_i|\mu_o(s, i|\phi), \Sigma_o(s, i|\phi)) \right) \right], \tag{19}$$

where $\mu_o(s, i|\phi), \Sigma_o(s, i|\phi)$ are the outputs of the projection network $h_\phi$.

**Off-policy Value Estimation**  The intuition of our work is to modify the sampling policy $\boldsymbol{\mu}$ in Equation (11), thus to derive a better optimization objective that can bring a smaller upper bound of the regret for the updated policy. However, term $A_{\boldsymbol{\pi}^k}$ is expected to be maintained which means that we want to estimate the advantage with the policy of the last round. For previous methods, it is easy to achieve because

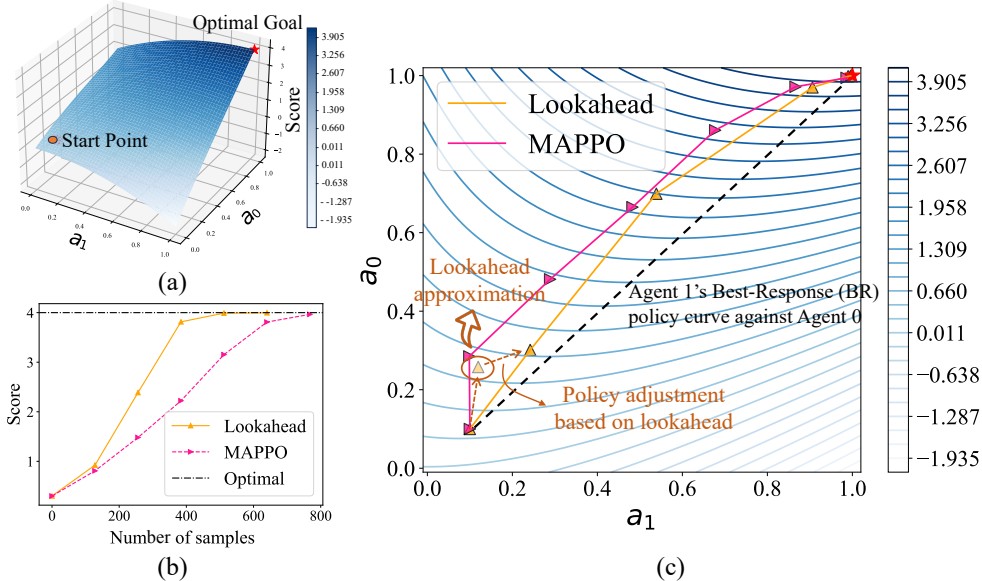

Figure 3: Visualization results on the toy environment. (a) Landscape of the two-variable function, where darker the color is, higher function score the region obtains; (b) Algorithm learning score curve; (c) Optimization process of Lookahead and MAPPO.

the trajectories are sampled by $\boldsymbol{\pi}^k$ and we can estimate the advantage directly. While in our algorithm design, the sampling policy is replaced with the lookahead policy $\tilde{\boldsymbol{\pi}}^{k+1}$, which means that we need to conduct off-policy estimation for $A_{\boldsymbol{\pi}^k}$. Specifically, we adopt the V-trace (Espeholt et al., 2018) trick to estimate $A_{\boldsymbol{\pi}^k}$ with the trajectories sampled by $\tilde{\boldsymbol{\pi}}^{k+1}$.

**Overall Flow of the Algorithm** Combining all the algorithmic design techniques that we have raised, we propose a practical algorithm that can enhance the underlying multi-agent policy gradient method. The overall flow of our algorithm has been presented in Algorithm 1. In line 5, we obtain the estimated future teammate policy $\tilde{\boldsymbol{\pi}}^{k+1}$ within the environment model, while in line 6 we utilize $\tilde{\boldsymbol{\pi}}^{k+1}$ to aid in actually updating the policy in the real environment. Besides, we update the environment model in line 4.

## 4 Experiments

In this section, we substantiate the efficacy of our proposed approach through empirical validation via experiments conducted on diverse benchmarks. These benchmarks include a toy environment, which serves to illustrate the algorithmic process of our approach, and two intricate cooperative multi-agent scenarios that provide practical validation of our approach's effectiveness. Specifically, we aim to utilize these experimental results to investigate the following questions: 1) How does our algorithm work and can we analyze the underlying mechanism through one simple task (Section 4.1)? 2) Can our algorithm actually enhance the cooperation learning in complex multi-agent cooperative tasks (Section 4.2)? 3) Does the phenomenon exhibited by our algorithm in complex cooperative tasks still align with our analysis (Section 4.3)?

### 4.1 Algorithm Analysis in Toy Environment

To visually reveal how our method works, we devised a toy environment involving a two-variable function optimization problem. As depicted in Figure 3(a), this problem comprises two agents with continuous action spaces in the range $[0, 1]$. Whenever the agents execute a joint action $[a_0, a_1]$, the environment yields reward: $R = a_0^3 - 2(a_0 - a_1)^2 + 3a_0$.

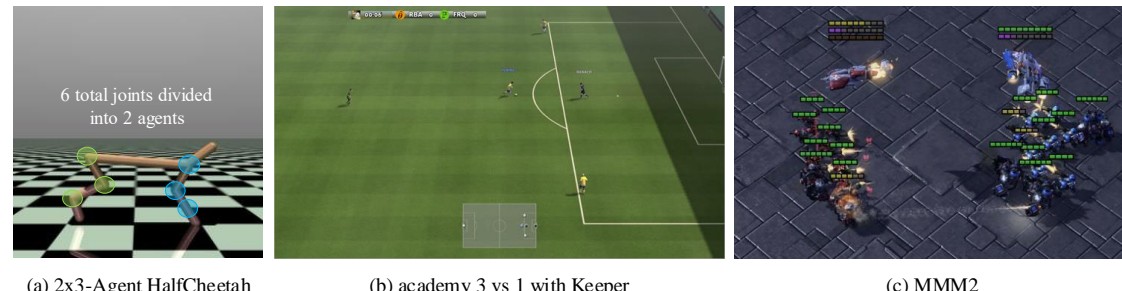

(a) 2x3-Agent HalfCheetah      (b) academy 3 vs 1 with Keeper      (c) MMM2

Figure 4: Some example task scenarios of Multi-Agent MuJoCo (MA-MuJoCo), Google Football Research (GRF), and StarCraft Multi-Agent Challenge (SMAC) environments.

Table 1: Evaluation results of various methods on MA-MuJoCo tasks, providing average scores across 5 seeds with standard errors. The highest score for each task scenario is **bolded** and the top-2 scores are marked in blue . The average rank denotes the average ranking across all task scenarios of each method.

| Algorithm | Ant 2x4 | Ant 4x2 | HalfCheetah 2x3 | HalfCheetah 3x2 | Walker2d 2x3 | Walker2d 3x2 | Average Rank |
|---|---|---|---|---|---|---|---|
| Lookahead | **3393.39**(336.98) | **2858.22**(540.76) | **3315.83**(346.46) | **3687.10**(304.98) | 2048.30(309.27) | 2670.71(123.74) | **1.33** |
| HAPPO | 2471.15(201.23) | 2120.18(168.48) | 2910.18(39.38) | 3016.20(80.90) | **2544.62**(272.91) | **2780.65**(76.25) | 2.00 |
| TAPPO | 2055.21(182.25) | 2503.62(207.66) | 2154.73(443.24) | 3487.52(711.65) | 1475.63(215.27) | 1445.56(350.34) | 3.67 |
| MAPPO | 1034.19(18.99) | 1002.15(18.30) | 2160.29(503.76) | 2350.32(477.29) | 1852.8(41.48) | 1812.54(139.44) | 4.33 |
| IPPO | 884.93(46.24) | 875.8(20.84) | 2652.04(635.31) | 2477.98(566.12) | 2021.89(67.64) | 1775.2(122.09) | 4.33 |
| MADDPG | 1866.08(9.77) | 1701.08(13.45) | 1553.54(251.6) | 1295.5(384.87) | 71.68(15.36) | 100.72(35.19) | 5.33 |

Specifically, we initialize the joint policy as $[0.1, 0.1]$ and both adopt the algorithms of MAPPO and our lookahead strategy to investigate how they converge to the optimal policy $[1, 1]$. As shown in Figure 3(c), without our lookahead strategy, agent 1 always shows a large gap from the Best-Response (BR) against the updated agent 0, revealing the phenomenon of "teammate delay". While our lookahead strategy can help agent 1 predict the updated policy of agent 0, leading to a shorter optimization path. From the learning curve in Figure 3(b), we also find that our lookahead strategy helps converge to the optimal policy using much fewer samples, enhancing the learning efficiency.

The algorithm analysis in this straightforward objective optimization task helps provide an intuitive explanation about the algorithm mechanism and motivation behind our approach. In the subsequent sections, we explore whether the proposed lookahead strategy can indeed enhance the cooperative learning in more complex task scenarios.

## 4.2 Main Results in Complex Cooperative Tasks

### 4.2.1 Experiment Setup

To investigate the effectiveness of our approach in more practical task scenarios, this section focuses on several prevalent cooperative benchmark environments, including continuous control tasks from Multi-Agent MuJoCo (MA-MuJoCo) (de Witt et al., 2020b) and Google Research Football (GRF) (Kurach et al., 2020) games with discrete action spaces. Besides, we also include experiments on StarCraft Multi-Agent Challenge (SMAC), which enables complex tasks involving larger team sizes. Examples of task scenarios from these three environments are illustrated in Figure 4. The introduction to each environment is provided below.

**Multi-Agent MuJoCo (MA-MuJoCo)** The MA-MuJoCo environment is built upon the MuJoCo physics engine to create realistic simulations for MARL research. In specific, MA-MuJoCo partitions the

Table 2: Evaluation results of various methods on GRF tasks. GRF 3vs1, CA(hard) and Corner are respectively short for maps of academy 3 vs 1 with keeper, academy counterattack hard and academy corner in GRF environment.

| Algorithm | GRF 3vs1 | GRF CA (hard) | GRF Corner | Average Rank |
|---|---|---|---|---|
| Lookahead | 0.82(0.02) | **0.50**(0.07) | **0.63**(0.03) | **1.33** |
| HAPPO | **0.86**(0.03) | 0.46(0.09) | 0.49(0.07) | 2.00 |
| TAPPO | 0.77(0.03) | 0.49(0.03) | 0.28(0.10) | 3.00 |
| MAPPO | 0.66(0.04) | 0.37(0.08) | 0.48(0.11) | 3.67 |
| CDS | 0.49(0.11) | 0.21(0.07) | 0.02(0.01) | 5.00 |

Table 3: Evaluation results of various methods on SMAC tasks, providing average scores across 5 seeds with standard errors.

| Algorithm | 3s5z | 5m_vs_6m | MMM2 | Average Rank |
|---|---|---|---|---|
| LookAhead | **94.2**(2.31) | **90.5**(2.51) | **90.6**(1.04) | **1.00** |
| HAPPO | 92.2(1.74) | 87.5(2.04) | 87.1(2.20) | 4.00 |
| TAPPO | 90.3(5.60) | 89.1(2.51) | 88.2(2.90) | 3.33 |
| MAPPO | 91.9(1.42) | 88.2(2.35) | 89.1(2.79) | 2.67 |
| IPPO | 91.9(2.51) | 87.5(2.93) | 87.5(2.51) | 3.67 |
| QMIX | 88.3(2.90) | 75.8(3.70) | 87.5(2.60) | 5.33 |

body graph in MuJoCo into disjoint sub-graphs, one for each agent, e.g., 2x4-Agent Ant means dividing the 8 joints in Ant into 2 agents, each controlling 4 joints.

**Google Research Football (GRF)** The Google Research Football (GRF) is a novel benchmark environment offering simulations of soccer matches, enabling the study of multi-agent behaviors and reinforcement learning. It introduces challenging cooperation learning tasks as it has the property of heterogeneity and sparse rewards.

**StarCraft Multi-Agent Challenge (SMAC)** The SMAC environment is also a popular benchmark for MARL, set in micromanagement scenarios with multi-agent and multi-unit battles. The goal is for the ally units to cooperate together to debate the enemy units. SMAC allows for the design of complex and challenging cooperative tasks due to the diversity of unit types and maps.

To thoroughly explore the cooperative performance that our approach can potentially bring about, we integrate our lookahead strategy with HAPPO, one of the current state-of-the-art multi-agent policy gradient algorithms, in all experiments of this section. For comparison, we select several popular multi-agent actor-critic algorithms as baselines. An opponent modeling approach, TAPPO, is also included, which learns teammate representations to incorporate additional policy conditions like in previous methods (Papoudakis & Albrecht, 2020; Cao et al., 2023). We adopt this baseline to contrast our approach with traditional opponent modeling approaches in mitigating non-stationarity issue arising from teammate co-learning. Moreover, in the GRF environment, we add one additional baseline CDS (Li et al., 2021), a value-based MARL algorithm designed specifically for solving GRF games, for a more comprehensive comparison. More details about baselines can be found in Appendix B.1.

### 4.2.2 Results Analysis

**MA-MuJoCo**   As shown in Table 1, in multiple task scenarios of MA-MuJoCo, our approach Lookahead has achieved superior cooperative performance compared to other baseline algorithms. For tasks of Ant and HalfCheetah, our approach has consistently achieved the highest scores across all methods. These results imply that in these task scenarios, through predicting the potential future policies of other agents controlling their respective joints, our algorithm can help agents learn to manipulate their own joints in coordination with other agents better, finally enhancing the cooperative performance. Despite not achieving the highest score, our approach also attains top-2 performance in the Walker2d scenarios. We hypothesize that the slight performance loss might be due to the nature of the Walker2d task, which requires not only proficient walking but also maintaining the balance of the mechanical legs at all times. The possible failure to maintain the legs' balance may pose great challenges to the learning of the environment model, consequently having a negative impact on the performance of our algorithm. Actually, the selection of environment model learning methods is orthogonal to our algorithm. In the future, we will consider designing better model learning methods to further enhance the performance of our approach.

**Google Research Football (GRF)**   Similar to the results in MA-MuJoCo, the results in Table 2 demonstrate that our approach also achieves superior performance in GRF problems. In particular, our approach is the only one that attains a success rate exceeding 60% in the academy corner scenario, achieving a notable improvement of over 25% compared to the second-best algorithm. Moreover, the average rank of our approach also stands out as the best in the GRF environment, the same as that in MA-MuJoCo.

**StraCraft Multi-Agent Challenge (SMAC)**   The results in Table 3 highlight the win rate performance of different approaches in SMAC, where our approach consistently achieves the best performance across three different task scenarios. These results, especially those on MMM2 task which involves ten ally agents, demonstrate the ability of our approach to adapt to more complex cooperative tasks, supporting the generality and adaptability of our approach.

**Ablation Study**   The comparison in Tables 1 to 3 between the Lookahead and HAPPO algorithms on both benchmark environments can be seen as ablation study to assess the impact of our introduced lookahead strategy. Across the majority of task scenarios, the incorporation of our lookahead strategy results in enhanced cooperative performance compared to the original HAPPO algorithm, e.g., exceeding the second-best algorithm by around 1000 points in 2x4-Agent Ant, which effectively validates the efficacy of the proposed lookahead strategy.

### 4.3 Analysis of Lookahead Policy in Complex Problems

While we have analyzed the algorithmic mechanism in a toy environment, applying the algorithm in complex tasks is more intricate due to the involvement of model learning. In this section, we measure the policy distances to investigate whether our lookahead approximation still provides right direction information in MA-MuJoCo. In specific, we compute the difference between $D_{\mathrm{KL}}(\boldsymbol{\pi}^{k+1}, \boldsymbol{\pi}^k)$ and $D_{\mathrm{KL}}(\boldsymbol{\pi}^{k+1}, \tilde{\boldsymbol{\pi}}^{k+1})$, which equals to $D_{\mathrm{KL}}(\boldsymbol{\pi}^{k+1}, \boldsymbol{\pi}^k) - D_{\mathrm{KL}}(\boldsymbol{\pi}^{k+1}, \tilde{\boldsymbol{\pi}}^{k+1})$. As we can see from Figure 5 that this metric consistently keep positive throughout the training process, the results reveal that $\tilde{\boldsymbol{\pi}}^{k+1}$ is actually closer to the future teammate policy $\boldsymbol{\pi}^{k+1}$, forming a relatively good approximation. These results support our motivation and more analytical experimental results can be found in Appendix C.

## 5   Related Work

The related work of this paper mainly covers three aspects: multi-agent policy gradient, multi-agent model learning, and opponent modeling. Below, we provide the introduction to related works in these three aspects respectively. Among them, opponent modeling approaches are typically proposed to address the general nonstationarity issue in multi-agent systems, considering the dynamic policy changes of both teammates and opponents. While related, our identified "teammate delay" problem is particularly pronounced in on-policy MARL algorithms for cooperative settings, which is an issue more related to algorithm design.

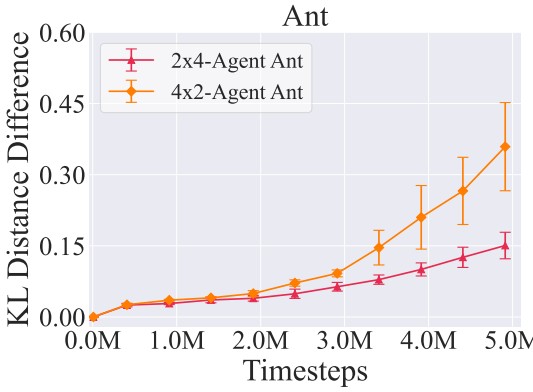 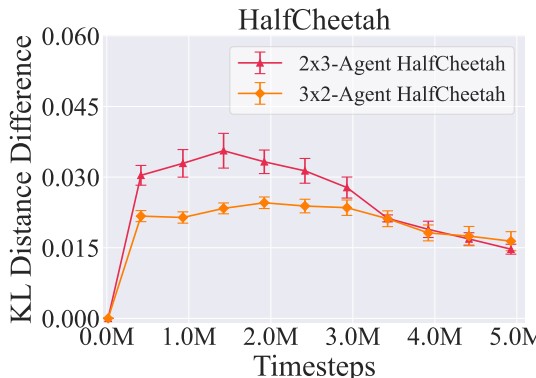

Figure 5: Measuring policy distances in Multi-Agent MuJoCo environments. The y-axis calculates the difference in distances from $\boldsymbol{\pi}^k$ and $\tilde{\boldsymbol{\pi}}^{k+1}$ to $\boldsymbol{\pi}^{k+1}$, i.e., $D_{\mathrm{KL}}(\boldsymbol{\pi}^{k+1}, \boldsymbol{\pi}^k) - D_{\mathrm{KL}}(\boldsymbol{\pi}^{k+1}, \tilde{\boldsymbol{\pi}}^{k+1})$, and the plot illustrates the changes in this metric throughout the policy training process.

**Multi-Agent Policy Gradient** The multi-agent policy gradient algorithms hold better convergence stability compared to the value-based algorithms, and they provide the ability to handle continuous control tasks. IA2C (Chu et al., 2019) introduces the A2C method to the multi-agent setting, and adopts an independent learning paradigm. Subsequently, COMA (Foerster et al., 2018b) proposes the paradigm of centralized critic with decentralized actors, which tries to conduct credit-assignment for each agent via introducing a counterfactual baseline. MAAC (Iqbal & Sha, 2019) and DOP (Wang et al., 2020) respectively improve the policy gradient methods by introducing the attention mechanism to the critic network and conducting value decomposition for the centralized critic. On the other hand, IPPO (de Witt et al., 2020a) and MAPPO (Yu et al., 2022a) extend the trust-region policy optimization scheme to the multi-agent setting, and obtain remarkable performance. However, all the policy gradient methods above directly optimize the agent policy associated with the current teammates, and may suffer from the "teammate delay" issue. Recently, HAPPO (Kuba et al., 2021) introduces sequential update scheme to the multi-agent policy gradient algorithm, which considers the mutual influences between different agents' policy update. Nevertheless, it adopts importance sampling technique which suffers from high variance, and it is orthogonal to our algorithm. More discussion is provided in Appendix B.2. Besides, there exist algorithms introducing deterministic policy gradient to the multi-agent setting (Lowe et al., 2017), while we mainly consider stochastic policy gradient methods here.

**Multi-Agent Model Learning** Model-based reinforcement learning enjoys higher sample efficiency. However, multi-agent model learning faces significant challenges due to the exponential growth of the state-action space and the non-stationary in multi-agent scenarios. Adopting the Dreamer (Hafner et al., 2019) architecture, MAMBA (Egorov & Shpilman, 2022) sustains a world model for each agent with necessary communication, thus to scale gracefully with the number of agents. Another work (Mahajan et al., 2021) shows utilizing tensor decomposition in multi-agent model learning can significantly improve the sample efficiency when the environment transition and reward functions are of low CP-rank. Krupnik (Krupnik et al., 2020) adopts generative models to learn a multi-step world model which can consider the delayed effects of the previous actions. Besides, considering the characteristics of multi-agent settings, AORPO (Zhang et al., 2021) and CTRL (Park et al., 2019) incorporate the opponent modeling into the model learning in order to roll-out opponent-wise trajectories. When the environment model has been obtained, dyna-style algorithms (Zhang et al., 2022; Willemsen et al., 2021) conduct data augmentation to enhance policy learning. MBVD (Xu et al., 2022) evaluates the current state value via imagining future states within the model. Han (Han et al., 2022) conducts credit assignment by computing the shapley value (Winter, 2002) using the samples rolled out in the model. In the future, a promising direction is to use generative world models for multi-agent environment modeling, enhancing the learning of multi-agent policies, as these generative world models have already shown considerable potential (Hu et al., 2023; Bruce et al., 2024; Qiao et al., 2024).

**Opponent Modeling**   Opponent modeling is a well-studied topic in the field of MARL. Some previous works utilize opponent modeling to alleviate the non-stationarity issue in MARL. Among them, some works (Hong et al., 2018; Papoudakis & Albrecht, 2020; Xie et al., 2021; Cao et al., 2023) involve utilizing the opponent representations as additional inputs to the policy network, thereby enhancing the policy learning. While AMS-A3C and AFS-A3C (Hernandez-Leal et al., 2019) treat the opponent modeling as an auxiliary task to guide the network optimization. Besides, another series of works assume that opponents are uncertain or may change, and they aim to help recognize and adapt to the opponents. DPN-BPR+ (Zheng et al., 2018) and MBOM (Yu et al., 2022b) estimate the most probable types of opponents from a statistical perspective, while Fastap (Zhang et al., 2023) further considers that the changes of teammates may happen within one episode and learn a instantaneous representation to achieve fast recognition of teammate changes. Moreover, there exist other series of works (Foerster et al., 2018a; Willi et al., 2022; Lu et al., 2022) that propose a better update operator for general-sum games by modeling the influences of agents' policies on the other agent. However, these methods are limited to two-player simple problems, while our work focuses on complex cooperative tasks.

## 6   Closing Remarks

This paper introduces a pioneering approach to enhance cooperative MARL by anticipating future teammate policies. Alleviating the prevalent issue of "teammate delay", our proposed lookahead strategy bridges the gap between the learning objective and the real evaluation scenario, significantly boosting the learning efficiency. Through seamless integration with existing gradient-based MARL methods, our approach surpasses state-of-the-art algorithms, exhibiting good performance in complex cooperative multi-agent benchmarks. Currently, our method mainly relies on the environment model to predict the future teammates. Thus, the practical algorithm performance is to some extent limited by the model learning error. How to better estimate the future teammates and whether there exist other ways to harness the predicted information of future teammates deserve further investigation. We believe researches in this topic can bring great advancement in the MARL domain.

## Acknowledgments

This work is supported by the National Science Foundation of China (62276126, 62495093, 62250069) and the Natural Science Foundation of Jiangsu (BK2024119, BK20243039).

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

# Appendix

## A Notations and Theoretical Analysis

### A.1 Notations

In Table 4, we list the main notations in our paper.

Table 4: Notation list.

| Symbol | Meaning |
| --- | --- |
| $\mathcal{S}, s$ | $\mathcal{S}$ denotes the state space for either the single-agent problem or multi-agent problem, while $s \in \mathcal{S}$ is an instance of the state. |
| $\mathcal{A}, a$ | $\mathcal{A}$ denotes the action space for the single-agent problem, while $a$ is an instance of the action. |
| $N$ | Number of the agents in multi-agent problems. |
| $\boldsymbol{\mathcal{A}}, \{\mathcal{A}_i\}_{i=1}^N$ | $\boldsymbol{\mathcal{A}}$ is the joint action space for the multi-agent problem, $\mathcal{A}_i$ is the action space for agent $i$. |
| $\boldsymbol{a}, \{a_i\}_{i=1}^N$ | $\boldsymbol{a} = [a_1, a_2, \cdots, a_N]$ is an instance of the joint action, where $a_i$ is the action for agent $i$. |
| $\mathcal{P}$ | Transition function for either the single-agent problem or multi-agent problem. |
| $\mathcal{R}$ | Reward function for either the single-agent problem or multi-agent problem. |
| $\gamma$ | Discount factor. |
| $\pi$ | $\pi$ denotes the policy for single-agent problems, where $\pi(a|s)$ means the probability of taking action $a$ under state $s$. |
| $\boldsymbol{\pi}, \{\pi_i\}_{i=1}^N$ | $\boldsymbol{\pi}$ denotes the joint policy for multi-agent problems, while $\pi_i$ indicates the policy for agent $i$. $\boldsymbol{\pi}(\boldsymbol{a}|s) = \prod_{i=1}^N \pi_i(a_i|s)$ means the probability of taking action $\boldsymbol{a}$ under state $s$. |
| $\boldsymbol{\pi}^k$ | The obtained joint policy after the $k$-th round of policy update. |
| $\eta(\boldsymbol{\pi})$ | The discounted return of joint policy $\boldsymbol{\pi}$ in multi-agent problems, i.e., $\eta(\boldsymbol{\pi}) = \mathbb{E}_{\boldsymbol{\pi}} \left[\sum_{t=0}^\infty \gamma^t r_t\right]$. |
| $d_{\boldsymbol{\pi}}(s),\ d_{\boldsymbol{\pi}}(s, \boldsymbol{a})$ | The stationary state visitation distribution and state-action visitation distribution when given the fixed joint policy $\boldsymbol{\pi}$. |
| $\rho_{\boldsymbol{\pi}}(s)$ | The unnormalized stationary state distribution derived by the joint policy $\boldsymbol{\pi}$. Specifically, $\rho_{\boldsymbol{\pi}}(s) = \frac{1}{1-\gamma} d_{\boldsymbol{\pi}}(s)$. |
| $Q_{\pi^k}(s, a), Q_{\boldsymbol{\pi}^k}(s, \boldsymbol{a})$ | $Q_{\pi^k}(s, a)$ represents the Q-function in single-agent problems, defined as the expected cumulative reward obtained by taking action $a$ in state $s$ and then following policy $\pi^k$ thereafter, i.e., $Q_{\pi^k}(s, a) = \mathbb{E}_{\pi^k} \left[\sum_{t=0}^\infty \gamma^t r_t | s_0 = s, a_0 = a\right]$. $Q_{\boldsymbol{\pi}^k}(s, \boldsymbol{a})$ is for multi-agent problems. |
| $V_{\pi^k}(s), V_{\boldsymbol{\pi}^k}(s)$ | $V_{\pi^k}(s)$ represents the state value function in single-agent problems, indicating the expected cumulative reward starting from state $s$ and following policy $\pi$ thereafter, i.e., $V_{\pi^k}(s) = \mathbb{E}_{\pi^k} \left[\sum_{t=0}^\infty \gamma^t r_t | s_0 = s\right]$. $V_{\boldsymbol{\pi}^k}(s)$ is for multi-agent problems. |
| $\alpha_i$ | $\alpha_i = \max_s D_{\mathrm{TV}}\left(\pi_i^{k+1}(\cdot|s) \| \mu_i(\cdot|s)\right)$ is utilized to denote the distance between the sampling policy and the updated policy (the policy at the next round). |

## A.2  Proofs of Main Theoretical Results

In this section, we provide the proofs of the main theoretical results in our paper. In specific, we begin by outlining the primary theoretical results below, followed by their respective proofs one-by-one. Among them, Lemma 1, Theorem 1, and Theorem 2 are introduced in the main text, while Theorem 3 is introduced in the supplementary discussion in Appendix A.3.

**Statement 1** *In previous multi-agent policy gradient methods, the learning objective for each agent in multi-agent policy gradient methods is typically defined as:*

$$J_i(\pi_i|\{\pi_j^k\}_{j\neq i}) = \eta(\boldsymbol{\pi}^k) + \sum_{s\in\mathcal{S}} \rho_{\boldsymbol{\pi}^k}(s)\left[\sum_{\boldsymbol{a}\in\mathcal{A}} \pi_i(a_i|s)\prod_{j\neq i}\pi_j^k(a_j|s)A_{\boldsymbol{\pi}^k}(s,\boldsymbol{a})\right], \ i\in\{1,2,\cdots,N\}. \tag{20}$$

**Lemma 1** *Assume that we update the joint policy $\boldsymbol{\pi}^k$ to $\boldsymbol{\pi}^{k+1}$ with sampling policy $\boldsymbol{\mu}$. Given the measurement of distance between sampling policy $\boldsymbol{\mu}$ and the updated policy $\boldsymbol{\pi}^{k+1}$ as $\alpha_i = \max_s D_{\mathrm{TV}}\left(\pi_i^{k+1}(\cdot|s)\|\mu_i(\cdot|s)\right)$, we have:*

$$|J(\boldsymbol{\pi}^{k+1}, \boldsymbol{\mu}) - \eta(\boldsymbol{\pi}^{k+1})| \leq \frac{4\epsilon\gamma}{(1-\gamma)^2}\left(\sum_{i=1}^{N}\alpha_i\right)^2 + \frac{2\epsilon(N-1)}{N}\sum_{i=1}^{N}\alpha_i, \tag{21}$$

*where $\gamma$ is the discount factor and $\epsilon = \max_{s,\boldsymbol{a}}|A_{\boldsymbol{\pi}^k}(s,\boldsymbol{a})|$.*

**Corollary 1** *Suppose that we update joint policy $\boldsymbol{\pi}^k$ to $\boldsymbol{\pi}^{k+1}$ with sampling policy $\boldsymbol{\mu}$, then the regret of the updated joint policy $\boldsymbol{\pi}^{k+1}$ has the following upper bound:*

$$\eta(\boldsymbol{\pi}^*) - \eta(\boldsymbol{\pi}^{k+1}) \leq \eta(\boldsymbol{\pi}^*) - J(\boldsymbol{\pi}^{k+1}, \boldsymbol{\mu}) + \frac{4\epsilon\gamma}{(1-\gamma)^2}\left(\sum_{i=1}^{N}\alpha_i\right)^2 + \frac{2\epsilon(N-1)}{N}\sum_{i=1}^{N}\alpha_i. \tag{22}$$

**Theorem 1** *Suppose the sampling policy $\boldsymbol{\mu}^*$ can derive the same updated policy, which means that $\boldsymbol{\mu}^* = \arg\max_{\boldsymbol{\pi}} J(\boldsymbol{\pi}, \boldsymbol{\mu}^*)$. If it exists, it will be the solution of the following bi-level optimization problem:*

$$\arg\min_{\boldsymbol{\mu}} D_{\mathrm{KL}}(\boldsymbol{\mu}\|\boldsymbol{\pi}^{k+1}), \ s.t. \ \boldsymbol{\pi}^{k+1} = \arg\max_{\boldsymbol{\pi}} J(\boldsymbol{\pi}, \boldsymbol{\mu}). \tag{23}$$

**Theorem 2** *Let $\phi$ be the ego-max-operator [2]. We suppose that $\boldsymbol{\mu}^*$ denotes the lookahead policy which means that it can derive the same updated policy, i.e., $\boldsymbol{\mu}^* = \psi(\boldsymbol{\mu}^*, \boldsymbol{\pi}^k)$; and $\boldsymbol{\pi}' = \psi(\boldsymbol{\pi}^k, \boldsymbol{\pi}^k)$ denotes the updated policy when using $\boldsymbol{\pi}^k$ as the sampling policy. We express the trust region as that $D_{\mathrm{TV}}\left(\pi_i'(\cdot|s)\|\pi_i^k(\cdot|s)\right) \leq \beta_i$. In this case, when $\beta_i \leq \frac{\sum_{s\in\mathcal{S}}\rho_{\boldsymbol{\mu}^*}(s)\phi(s, A_{\boldsymbol{\pi}^k})}{2\sum_{s\in\mathcal{S}}\rho_{\boldsymbol{\pi}^k}(s)\phi(s, |A_{\boldsymbol{\pi}^k}|)}$, we have $J(\boldsymbol{\mu}^*, \boldsymbol{\mu}^*) \geq J(\boldsymbol{\pi}', \boldsymbol{\pi}^k)$.*

**Below are proofs.**

**Statement 1** *In previous multi-agent policy gradient methods, the learning objective for each agent is typically defined as:*

$$J_i(\pi_i|\{\pi_j^k\}_{j\neq i}) = \eta(\boldsymbol{\pi}^k) + \sum_{s\in\mathcal{S}} \rho_{\boldsymbol{\pi}^k}(s)\left[\sum_{\boldsymbol{a}\in\mathcal{A}} \pi_i(a_i|s)\prod_{j\neq i}\pi_j^k(a_j|s)A_{\boldsymbol{\pi}^k}(s,\boldsymbol{a})\right], \ i\in\{1,2,\cdots,N\}. \tag{24}$$

*Proof.* Based on the previous literature, we have already know that for single-agent setting, the learning objective is typically defined as:

$$J(\pi) = \eta(\boldsymbol{\pi}^k) + \sum_{s\in\mathcal{S}} \rho_{\pi^k}(s)\left[\sum_{a\in\mathcal{A}} \pi(a|s)A_{\pi^k}(s,a)\right]. \tag{25}$$

---

[2]Assuming $f$ is a function defined over the state and joint action space, the ego-max-operator $\phi$ is defined as $\phi(f,s) = \frac{1}{N}\sum_{i=1}^{N}\max_{a_i\in\mathcal{A}}\sum_{a_{-i}\in\mathcal{A}}\prod_{j\neq i}\pi_j^k(a_j|s)f(s,\boldsymbol{a})$.

However, since $\pi$ is the policy we are optimizing, we can not obtain the training data belonging to its distribution perfectly. Thus, in practice, the objective is typically transformed into:

$$J(\pi) = \eta(\pi^k) + \sum_{s \in \mathcal{S}} \rho_{\pi^k}(s) \left[ \sum_{a \in \mathcal{A}} \pi^k(a|s) \frac{\pi(a|s)}{\pi^k(a|s)} A_{\pi^k}(s,a) \right], \tag{26}$$

$$= \eta(\pi^k) + \mathbb{E}_{(s,a) \sim \pi^k} \left[ \frac{\pi(a|s)}{\pi^k(a|s)} A_{\pi^k}(s,a) \right]. \tag{27}$$

This allows us to optimize the training objective through sampling trajectories using $\pi^k$, which is the policy at the last round. For multi-agent cases, the situation is similar. The previous multi-agent policy gradient methods typically sample data utilizing the joint policy $\boldsymbol{\pi}^k$ to sample training data and the ideal goal is to optimize the following objective:

$$J(\boldsymbol{\pi}) = \eta(\boldsymbol{\pi}^k) + \sum_{s \in \mathcal{S}} \rho_{\boldsymbol{\pi}^k}(s) \left[ \sum_{\boldsymbol{a} \in \mathcal{A}} \boldsymbol{\pi}^k(\boldsymbol{a}|s) \frac{\boldsymbol{\pi}(\boldsymbol{a}|s)}{\boldsymbol{\pi}^k(\boldsymbol{a}|s)} A_{\boldsymbol{\pi}^k}(s,\boldsymbol{a}) \right], \tag{28}$$

$$= \eta(\boldsymbol{\pi}^k) + \mathbb{E}_{(s,\boldsymbol{a}) \sim \boldsymbol{\pi}^k} \left[ \frac{\boldsymbol{\pi}(\boldsymbol{a}|s)}{\boldsymbol{\pi}^k(\boldsymbol{a}|s)} A_{\boldsymbol{\pi}^k}(s,\boldsymbol{a}) \right]. \tag{29}$$

However, since the candidate space of joint policy $\boldsymbol{\pi}$ is huge and it is hard to optimize directly. The previous multi-policy gradient methods actually conduct a trade-off, and optimize a decomposed objective for each agent (Yu et al., 2022a):

$$J_i(\pi_i) = \eta(\boldsymbol{\pi}^k) + \mathbb{E}_{(s,\boldsymbol{a}) \sim \boldsymbol{\pi}^k} \left[ \frac{\pi_i(a_i|s)}{\pi_i^k(a_i|s)} A_{\boldsymbol{\pi}^k}(s,\boldsymbol{a}) \right], \ i \in \{1, 2, \cdots, N\}. \tag{30}$$

Equation (30) can actually expressed as:

$$J_i(\pi_i|\{\pi_j^k\}_{j \neq i}) = \eta(\boldsymbol{\pi}^k) + \sum_{s \in \mathcal{S}} \rho_{\boldsymbol{\pi}^k}(s) \left[ \sum_{\boldsymbol{a} \in \mathcal{A}} \pi_i(a_i|s) \prod_{j \neq i} \pi_j^k(a_j|s) A_{\boldsymbol{\pi}^k}(s,\boldsymbol{a}) \right], \ i \in \{1, 2, \cdots, N\}. \tag{31}$$

$\square$

**Lemma 1** *Assume that we update the joint policy $\boldsymbol{\pi}^k$ to $\boldsymbol{\pi}^{k+1}$ with sampling policy $\boldsymbol{\mu}$. Given the measurement of distance between sampling policy $\boldsymbol{\mu}$ and the updated policy $\boldsymbol{\pi}^{k+1}$ as $\alpha_i = \max_s D_{\mathrm{TV}}\left(\pi_i^{k+1}(\cdot|s) \| \mu_i(\cdot|s)\right)$, we have:*

$$|J(\boldsymbol{\pi}^{k+1}, \boldsymbol{\mu}) - \eta(\boldsymbol{\pi}^{k+1})| \leq \frac{4\epsilon\gamma}{(1-\gamma)^2} \left( \sum_{i=1}^{N} \alpha_i \right)^2 + \frac{2\epsilon(N-1)}{N} \sum_{i=1}^{N} \alpha_i, \tag{32}$$

*where $\gamma$ is the discount factor and $\epsilon = \max_{s,\boldsymbol{a}} |A_{\boldsymbol{\pi}^k}(s,\boldsymbol{a})|$.*

*Proof.* Firstly, according to the *performance difference lemma* (Kakade & Langford, 2002), we have:

$$
\begin{aligned}
&\eta(\boldsymbol{\pi}^{k+1}) - J(\boldsymbol{\pi}^{k+1}, \boldsymbol{\mu}) \\
&= \sum_{s \in \mathcal{S}} \rho_{\boldsymbol{\pi}^{k+1}}(s) \sum_{\boldsymbol{a} \in \mathcal{A}} \boldsymbol{\pi}^{k+1}(\boldsymbol{a}|s) A_{\boldsymbol{\pi}^k}(s, \boldsymbol{a}) \\
&\qquad\qquad\qquad\qquad - \sum_{s \in \mathcal{S}} \rho_{\boldsymbol{\mu}}(s) \left[ \frac{1}{N} \sum_{i=1}^{N} \sum_{\boldsymbol{a} \in \mathcal{A}} \pi_i(a_i|s) \prod_{j \neq i} \mu_j(a_j|s) A_{\boldsymbol{\pi}^k}(s, \boldsymbol{a}) \right] \\
&= \sum_{s \in \mathcal{S}} \rho_{\boldsymbol{\pi}^{k+1}}(s) \sum_{\boldsymbol{a} \in \mathcal{A}} \boldsymbol{\pi}^{k+1}(\boldsymbol{a}|s) A_{\boldsymbol{\pi}^k}(s, \boldsymbol{a}) - \sum_{s \in \mathcal{S}} \rho_{\boldsymbol{\mu}}(s) \sum_{\boldsymbol{a} \in \mathcal{A}} \boldsymbol{\pi}^{k+1}(\boldsymbol{a}|s) A_{\boldsymbol{\pi}^k}(s, \boldsymbol{a}) \\
&\qquad\qquad + \sum_{s \in \mathcal{S}} \rho_{\boldsymbol{\mu}}(s) \sum_{\boldsymbol{a} \in \mathcal{A}} \boldsymbol{\pi}^{k+1}(\boldsymbol{a}|s) A_{\boldsymbol{\pi}^k}(s, \boldsymbol{a}) \\
&\qquad\qquad - \sum_{s \in \mathcal{S}} \rho_{\boldsymbol{\mu}}(s) \left[ \frac{1}{N} \sum_{i=1}^{N} \sum_{\boldsymbol{a} \in \mathcal{A}} \pi_i(a_i|s) \prod_{j \neq i} \mu_j(a_j|s) A_{\boldsymbol{\pi}^k}(s, \boldsymbol{a}) \right] \\
&\leq \left| \sum_{s \in \mathcal{S}} \rho_{\boldsymbol{\pi}^{k+1}}(s) \sum_{\boldsymbol{a} \in \mathcal{A}} \boldsymbol{\pi}^{k+1}(\boldsymbol{a}|s) A_{\boldsymbol{\pi}^k}(s, \boldsymbol{a}) - \sum_{s \in \mathcal{S}} \rho_{\boldsymbol{\mu}}(s) \sum_{\boldsymbol{a} \in \mathcal{A}} \boldsymbol{\pi}^{k+1}(\boldsymbol{a}|s) A_{\boldsymbol{\pi}^k}(s, \boldsymbol{a}) \right| \\
&\qquad + \left| \sum_{s \in \mathcal{S}} \rho_{\boldsymbol{\mu}}(s) \sum_{\boldsymbol{a} \in \mathcal{A}} \boldsymbol{\pi}^{k+1}(\boldsymbol{a}|s) A_{\boldsymbol{\pi}^k}(s, \boldsymbol{a}) \right. \\
&\qquad\qquad \left. - \sum_{s \in \mathcal{S}} \rho_{\boldsymbol{\mu}}(s) \left[ \frac{1}{N} \sum_{i=1}^{N} \sum_{\boldsymbol{a} \in \mathcal{A}} \pi_i(a_i|s) \prod_{j \neq i} \mu_j(a_j|s) A_{\boldsymbol{\pi}^k}(s, \boldsymbol{a}) \right] \right| \\
&\overset{\text{(I)}}{\leq} \frac{4\epsilon\gamma}{(1-\gamma)^2} \alpha^2 + \left| \sum_{s \in \mathcal{S}} \rho_{\boldsymbol{\mu}}(s) \sum_{\boldsymbol{a} \in \mathcal{A}} \boldsymbol{\pi}^{k+1}(\boldsymbol{a}|s) A_{\boldsymbol{\pi}^k}(s, \boldsymbol{a}) \right. \\
&\qquad\qquad \left. - \sum_{s \in \mathcal{S}} \rho_{\boldsymbol{\mu}}(s) \left[ \frac{1}{N} \sum_{i=1}^{N} \sum_{\boldsymbol{a} \in \mathcal{A}} \pi_i(a_i|s) \prod_{j \neq i} \mu_j(a_j|s) A_{\boldsymbol{\pi}^k}(s, \boldsymbol{a}) \right] \right|,
\end{aligned}
\tag{33}
$$

where (I) holds because of the conclusion that has already been obtained in TRPO (Schulman et al., 2015) (see Theorem 1). Besides, we further have:

$$
\begin{aligned}
&\left| \sum_{s \in \mathcal{S}} \rho_{\boldsymbol{\mu}}(s) \sum_{\boldsymbol{a} \in \mathcal{A}} \boldsymbol{\pi}^{k+1}(\boldsymbol{a}|s) A_{\boldsymbol{\pi}^k}(s, \boldsymbol{a}) \right. \\
&\qquad\qquad \left. - \sum_{s \in \mathcal{S}} \rho_{\boldsymbol{\mu}}(s) \left[ \frac{1}{N} \sum_{i=1}^{N} \sum_{\boldsymbol{a} \in \mathcal{A}} \pi_i^{k+1}(a_i|s) \prod_{j \neq i} \mu_j(a_j|s) A_{\boldsymbol{\pi}^k}(s, \boldsymbol{a}) \right] \right| \\
&\leq \left| \epsilon \sum_{s \in \mathcal{S}} \rho_{\boldsymbol{\mu}}(s) \frac{1}{N} \sum_{i=1}^{N} \sum_{a_i \in \mathcal{A}_i} \pi_i^{k+1}(a_i|s) \sum_{a_{-i} \in \mathcal{A}_{-i}} \left( \prod_{j \neq i} \mu_j(a_j|s) - \prod_{j \neq i} \pi_j^{k+1}(a_j|s) \right) \right| \\
&\overset{\text{(II)}}{=} \epsilon \sum_{s \in \mathcal{S}} \rho_{\boldsymbol{\mu}}(s) \frac{2}{N} \sum_{i=1}^{N} \sum_{a_i \in \mathcal{A}_i} \pi_i^{k+1}(a_i|s) D_{\text{TV}} \left( \pi_{-i}^{k+1}(\cdot|s) \| \mu_{-i}(\cdot|s) \right) \\
&= \frac{2\epsilon}{N} \sum_{s \in \mathcal{S}} \rho_{\boldsymbol{\mu}}(s) \sum_{i=1}^{N} D_{\text{TV}} \left( \pi_{-i}^{k+1}(\cdot|s) \| \mu_{-i}(\cdot|s) \right) \\
&\leq \frac{2\epsilon}{N} \sum_{i=1}^{N} \max_{s} D_{\text{TV}} \left( \pi_{-i}^{k+1}(\cdot|s) \| \mu_{-i}(\cdot|s) \right),
\end{aligned}
\tag{34}
$$

where (II) holds according to the definition of TV distance. Thus, we finally have:

$$
\begin{aligned}
\eta(\boldsymbol{\pi}^{k+1}) - J(\boldsymbol{\pi}^{k+1}, \boldsymbol{\mu}) &\le \frac{4\epsilon\gamma}{(1-\gamma)^2}\alpha^2 + \frac{2\epsilon}{N}\sum_{i=1}^{N}\max_s D_{\mathrm{TV}}\left(\pi_{-i}^{k+1}(\cdot|s)\|\mu_{-i}(\cdot|s)\right) \\
&= \frac{4\epsilon\gamma}{(1-\gamma)^2}\alpha^2 + \frac{2\epsilon(N-1)}{N}\sum_{i=1}^{N}\alpha_i \\
&\overset{\text{(III)}}{\le} \underbrace{\frac{4\epsilon\gamma}{(1-\gamma)^2}\left(\sum_{i=1}^{N}\alpha_i\right)^2}_{(a)} + \underbrace{\frac{2\epsilon(N-1)}{N}\sum_{i=1}^{N}\alpha_i}_{(b)},
\end{aligned}
\tag{35}
$$

where (III) is because $\alpha \le \sum_{i=1}^{N}\alpha_i$. Specifically, term $(a)$ in the final upper bound is due to the state distribution mismatch of the training trajectories, while term $(b)$ reveals the impact of the "teammate delay" phenomenon on the learning objective. $\qquad\square$

**Corollary 1** *Suppose that we update joint policy $\boldsymbol{\pi}^k$ to $\boldsymbol{\pi}^{k+1}$ with sampling policy $\boldsymbol{\mu}$, then the regret of the updated joint policy $\boldsymbol{\pi}^{k+1}$ has the following upper bound:*

$$
\eta(\boldsymbol{\pi}^*) - \eta(\boldsymbol{\pi}^{k+1}) \le \eta(\boldsymbol{\pi}^*) - J(\boldsymbol{\pi}^{k+1}, \boldsymbol{\mu}) + \frac{4\epsilon\gamma}{(1-\gamma)^2}\left(\sum_{i=1}^{N}\alpha_i\right)^2 + \frac{2\epsilon(N-1)}{N}\sum_{i=1}^{N}\alpha_i.
\tag{36}
$$

*Proof.*

$$
\begin{aligned}
&\eta(\boldsymbol{\pi}^*) - \eta(\boldsymbol{\pi}^{k+1}) \\
&= \eta(\boldsymbol{\pi}^*) - J(\boldsymbol{\pi}^{k+1}, \boldsymbol{\mu}) + J(\boldsymbol{\pi}^{k+1}, \boldsymbol{\mu}) - \eta(\boldsymbol{\pi}^{k+1}) \\
&\le \eta(\boldsymbol{\pi}^*) - J(\boldsymbol{\pi}^{k+1}, \boldsymbol{\mu}) + \left| J(\boldsymbol{\pi}^{k+1}, \boldsymbol{\mu}) - \eta(\boldsymbol{\pi}^{k+1}) \right| \\
&\overset{\text{(IV)}}{\le} \eta(\boldsymbol{\pi}^*) - J(\boldsymbol{\pi}^{k+1}, \boldsymbol{\mu}) + \frac{4\epsilon\gamma}{(1-\gamma)^2}\left(\sum_{i=1}^{N}\alpha_i\right)^2 + \frac{2\epsilon(N-1)}{N}\sum_{i=1}^{N}\alpha_i,
\end{aligned}
\tag{37}
$$

where (IV) is obtained due to *Lemma 1*. $\qquad\square$

**Theorem 1** *Suppose the sampling policy $\boldsymbol{\mu}^*$ can derive the same updated policy, which means that $\boldsymbol{\mu}^* = \arg\max_{\boldsymbol{\pi}} J(\boldsymbol{\pi}, \boldsymbol{\mu}^*)$. If it exists, it will be the solution of the following bi-level optimization problem:*

$$
\arg\min_{\boldsymbol{\mu}} D_{\mathrm{KL}}(\boldsymbol{\mu}\|\boldsymbol{\pi}^{k+1}), \ \ s.t. \ \boldsymbol{\pi}^{k+1} = \arg\max_{\boldsymbol{\pi}} J(\boldsymbol{\pi}, \boldsymbol{\mu}).
\tag{38}
$$

*Proof.* This theorem implicitly conveys a twofold meaning. First, we will prove that $\boldsymbol{\mu}^*$ is one optimal solution of this optimization problem. Second, we will prove that each solution $\boldsymbol{\mu}'$ for this optimization problem can derive the same updated policy when serving as the sampling policy.

First, for $\boldsymbol{\mu} = \boldsymbol{\mu}^*$, we know that the corresponding $\boldsymbol{\pi}^{k+1} = \arg\max_{\boldsymbol{\pi}} J(\boldsymbol{\pi}, \boldsymbol{\mu})$ is also $\boldsymbol{\mu}^*$, due to the definition of $\boldsymbol{\mu}^*$. Then we have:

$$
D_{\mathrm{KL}}(\boldsymbol{\mu}\|\boldsymbol{\pi}^{k+1}) = D_{\mathrm{KL}}(\boldsymbol{\mu}^*\|\boldsymbol{\mu}^*) = 0
\tag{39}
$$

Moreover, since $D_{\mathrm{KL}}(\boldsymbol{\mu}\|\boldsymbol{\pi}^{k+1}) \ge 0$, we know that $\boldsymbol{\mu}^*$ achieves the lowest value of $D_{\mathrm{KL}}(\boldsymbol{\mu}\|\boldsymbol{\pi}^{k+1})$. Thus, $\boldsymbol{\mu}^*$ is one solution for this bi-level optimization problem.

Second, suppose $\boldsymbol{\mu}'$ is one solution for this bi-level optimization problem. Then according to the definition of this problem, we should have:

$$
D_{\mathrm{KL}}(\boldsymbol{\mu}'\|\arg\max_{\boldsymbol{\pi}} J(\boldsymbol{\pi}, \boldsymbol{\mu}')) \le D_{\mathrm{KL}}(\boldsymbol{\mu}^*\|\boldsymbol{\mu}^*) = 0,
\tag{40}
$$

It can be inferred that $D_{\mathrm{KL}}(\boldsymbol{\mu}'\|\arg\max_{\boldsymbol{\pi}} J(\boldsymbol{\pi}, \boldsymbol{\mu}')) = 0$, implying that $\arg\max_{\boldsymbol{\pi}} J(\boldsymbol{\pi}, \boldsymbol{\mu}')$ and $\boldsymbol{\mu}'$ have the same distributions. Thus we know $\boldsymbol{\mu}' = \arg\max_{\boldsymbol{\pi}} J(\boldsymbol{\pi}, \boldsymbol{\mu}')$. $\qquad\square$

**Theorem 2** *Let $\phi$ be the ego-max-operator. We suppose that $\boldsymbol{\mu}^*$ denotes the lookahead policy which means that it can derive the same updated policy, i.e., $\boldsymbol{\mu}^* = \psi(\boldsymbol{\mu}^*, \boldsymbol{\pi}^k)$; and $\boldsymbol{\pi}' = \psi(\boldsymbol{\pi}^k, \boldsymbol{\pi}^k)$ denotes the updated policy when using $\boldsymbol{\pi}^k$ as the sampling policy. We express the trust region as that $D_{\text{TV}}\left(\pi_i'(\cdot|s)\|\pi_i^k(\cdot|s)\right) \leq \beta_i$. In this case, when $\beta_i \leq \frac{\sum_{s\in\mathcal{S}} \rho_{\boldsymbol{\mu}^*}(s)\phi(s, A_{\boldsymbol{\pi}^k})}{2\sum_{s\in\mathcal{S}} \rho_{\boldsymbol{\pi}^k}(s)\phi(s, |A_{\boldsymbol{\pi}^k}|)}$, we have $J(\boldsymbol{\mu}^*, \boldsymbol{\mu}^*) \geq J(\boldsymbol{\pi}', \boldsymbol{\pi}^k)$.*

*Proof.* To begin with, for the updated policy $\boldsymbol{\pi}' = \psi(\boldsymbol{\pi}^k, \boldsymbol{\pi}^k)$ using $\boldsymbol{\pi}^k$ as the sampling policy, we have:

$$
\begin{aligned}
J(\boldsymbol{\pi}', \boldsymbol{\pi}^k) &= \eta(\boldsymbol{\pi}^k) + \sum_{s\in\mathcal{S}} \rho_{\boldsymbol{\pi}^k}(s) \left[ \frac{1}{N} \sum_{i=1}^{N} \sum_{a\in\mathcal{A}} \pi_i'(a_i|s) \prod_{j\neq i} \pi_j^k(a_j|s) A_{\boldsymbol{\pi}^k}(s, a) \right] \\
\implies & |J(\boldsymbol{\pi}', \boldsymbol{\pi}^k) - J(\boldsymbol{\pi}^k, \boldsymbol{\pi}^k)| \\
&\leq \sum_{s\in\mathcal{S}} \rho_{\boldsymbol{\pi}^k}(s) \left[ \frac{1}{N} \sum_{i=1}^{N} \sum_{a\in\mathcal{A}} \left| (\pi_i'(a_i|s) - \pi_i^k(a_i|s)) \prod_{j\neq i} \pi_j^k(a_j|s) A_{\boldsymbol{\pi}^k}(s, a) \right| \right] \\
&= \sum_{s\in\mathcal{S}} \frac{\rho_{\boldsymbol{\pi}^k}(s)}{N} \sum_{i=1}^{N} \sum_{a\in\mathcal{A}} |\pi_i'(a_i|s) - \pi_i^k(a_i|s)| \prod_{j\neq i} \pi_j^k(a_j|s) |A_{\boldsymbol{\pi}^k}(s, a)| \\
&\leq \sum_{s\in\mathcal{S}} \frac{2\rho_{\boldsymbol{\pi}^k}(s)}{N} \sum_{i=1}^{N} D_{\text{TV}}(\pi_i'(\cdot|s)\|\pi_i^k(\cdot|s)) \max_{a_i\in\mathcal{A}} \sum_{a_{-i}\in\mathcal{A}} \prod_{j\neq i} \pi_j^k(a_j|s) |A_{\boldsymbol{\pi}^k}(s, a)|.
\end{aligned}
\tag{41}
$$

We have trust region condition that $\forall s \in \mathcal{S}, D_{\text{TV}}(\pi_i'(\cdot|s)\|\pi_i^k(\cdot|s)) \leq \beta_i$. We assume $\beta_i$ has an upper bound $\zeta$ for each agent $i$, then we further have:

$$
\begin{aligned}
|J(\boldsymbol{\pi}', \boldsymbol{\pi}^k) - J(\boldsymbol{\pi}^k, \boldsymbol{\pi}^k)| &\leq 2\zeta \sum_{s\in\mathcal{S}} \rho_{\boldsymbol{\pi}^k}(s) \left[ \frac{1}{N} \sum_{i=1}^{N} \max_{a_i\in\mathcal{A}} \sum_{a_{-i}\in\mathcal{A}} \prod_{j\neq i} \pi_j^k(a_j|s) |A_{\boldsymbol{\pi}^k}(s, a)| \right] \\
\rightarrow J(\boldsymbol{\pi}', \boldsymbol{\pi}^k) &\leq J(\boldsymbol{\pi}^k, \boldsymbol{\pi}^k) + 2\zeta \sum_{s\in\mathcal{S}} \rho_{\boldsymbol{\pi}^k}(s) \left[ \frac{1}{N} \sum_{i=1}^{N} \max_{a_i\in\mathcal{A}} \sum_{a_{-i}\in\mathcal{A}} \prod_{j\neq i} \pi_j^k(a_j|s) |A_{\boldsymbol{\pi}^k}(s, a)| \right] \\
\rightarrow J(\boldsymbol{\pi}', \boldsymbol{\pi}^k) &\leq \eta(\boldsymbol{\pi}^k) + 2\zeta \sum_{s\in\mathcal{S}} \rho_{\boldsymbol{\pi}^k}(s) \left[ \frac{1}{N} \sum_{i=1}^{N} \max_{a_i\in\mathcal{A}} \sum_{a_{-i}\in\mathcal{A}} \prod_{j\neq i} \pi_j^k(a_j|s) |A_{\boldsymbol{\pi}^k}(s, a)| \right].
\end{aligned}
\tag{42}
$$

We know that $J(\boldsymbol{\mu}^*, \boldsymbol{\mu}^*) = \arg\max_{\boldsymbol{\pi}\in\text{Ball}(\boldsymbol{\mu}^*)} J(\boldsymbol{\pi}, \boldsymbol{\mu}^*)$, where $\text{Ball}(\boldsymbol{\mu}^*)$ means the trust region of $\boldsymbol{\mu}^*$. Further, this optimization problem means that:

$$
\arg\max_{\boldsymbol{\pi}\in\text{Ball}(\mu)} J(\boldsymbol{\pi}, \boldsymbol{\mu}^*) = \eta(\boldsymbol{\pi}^k) + \sum_{s\in\mathcal{S}} \rho_{\boldsymbol{\mu}^*}(s) \left[ \frac{1}{N} \sum_{i=1}^{N} \sum_{a\in\mathcal{A}} \pi_i(a_i|s) \prod_{j\neq i} \mu_j^*(a_j|s) A_{\boldsymbol{\pi}^k}(s, a) \right].
\tag{43}
$$

Thus, we are actually to optimize $\frac{1}{N} \sum_{i=1}^{N} \sum_{a\in\mathcal{A}} \pi_i(a_i|s) \prod_{j\neq i} \mu_j^*(a_j|s) A_{\boldsymbol{\pi}^k}(s, a)$ for each $s \in \mathcal{S}$; and when we find the optimized results are actually $\mu$, it means that for each $s \in \mathcal{S}$, $\mu(\cdot|s)$ is a nash equilibrium for the cooperative game where the utility of action $a$ is defined as $A_{\boldsymbol{\pi}^k}(s, a)$. With proper updating scheme, it is reasonable that we can obtain equilibrium that satisfies:

$$
\begin{aligned}
\sum_{a\in\mathcal{A}} \mu_i^*(a_i|s) \prod_{j\neq i} \mu_j^*(a_j|s) A_{\boldsymbol{\pi}^k}(s, a) &= \sum_{a_i\in\mathcal{A}} \mu_i^*(a_i|s) \sum_{a_{-i}\in\mathcal{A}} \prod_{j\neq i} \mu_j^*(a_j|s) A_{\boldsymbol{\pi}^k}(s, a) \\
&\geq \max_{a_i\in\mathcal{A}} \sum_{a_{-i}\in\mathcal{A}} \prod_{j\neq i} \pi_j^k(a_j|s) A_{\boldsymbol{\pi}^k}(s, a).
\end{aligned}
\tag{44}
$$

Thus we have that:

$$
\begin{aligned}
J(\boldsymbol{\mu}^*, \boldsymbol{\mu}^*) &= \eta(\boldsymbol{\pi}^k) + \sum_{s \in \mathcal{S}} \rho_{\boldsymbol{\mu}^*}(s) \left[ \frac{1}{N} \sum_{i=1}^{N} \sum_{a \in \mathcal{A}} \mu_i^*(a_i|s) \prod_{j \neq i} \mu_j^*(a_j|s) A_{\boldsymbol{\pi}^k}(s, a) \right] \\
&\geq \eta(\boldsymbol{\pi}^k) + \sum_{s \in \mathcal{S}} \rho_{\boldsymbol{\mu}^*}(s) \left[ \frac{1}{N} \sum_{i=1}^{N} \max_{a_i \in \mathcal{A}} \sum_{a_{-i} \in \mathcal{A}} \prod_{j \neq i} \pi_j^k(a_j|s) A_{\boldsymbol{\pi}^k}(s, a) \right].
\end{aligned}
\tag{45}
$$

We define $\phi(f, s) = \frac{1}{N} \sum_{i=1}^{N} \max_{a_i \in \mathcal{A}} \sum_{a_{-i} \in \mathcal{A}} \prod_{j \neq i} \pi_j^k(a_j|s) f(s, \boldsymbol{a})$. Then when $\beta_i \leq \zeta \leq \frac{\sum_{s \in \mathcal{S}} \rho_{\boldsymbol{\mu}^*}(s)\phi(s, A_{\boldsymbol{\pi}^k})}{2 \sum_{s \in \mathcal{S}} \rho_{\boldsymbol{\pi}^k}(s)\phi(s, |A_{\boldsymbol{\pi}^k}|)}$, according to Equation (42) and Equation (45), it is obvious that $J(\boldsymbol{\mu}^*, \boldsymbol{\mu}^*) \geq J(\boldsymbol{\pi}', \boldsymbol{\pi}^k)$. $\qquad\square$

### A.3 Extra Analysis on Upper Bound

Theorem 1 has told us that when the extra term ($c$) in the upper bound disappears when we train the agents with future teammate information, which motivates us to predict future teammates. However, we still retrain a question whether eliminating term ($c$) can indeed reduce the overall upper bound. For this question, one potential risk is that eliminating term ($c$) might influence the optimization of $-J(\boldsymbol{\pi}, \boldsymbol{\mu})$, thus making the second term $-J(\boldsymbol{\pi}^{k+1}, \boldsymbol{\mu})$ larger. To solve this concern, we prove that under certain conditions it is at least better than the previous algorithms, as described below.

**Theorem 2** *To begin with, we introduce the policy update operator $\psi$ [3] and ego-max-operator $\phi$ [4]. We suppose that $\boldsymbol{\mu}^*$ denotes the lookahead policy which means that it can derive the same updated policy, i.e., $\boldsymbol{\mu}^* = \psi(\boldsymbol{\mu}^*, \boldsymbol{\pi}^k)$; and $\boldsymbol{\pi}' = \psi(\boldsymbol{\pi}^k, \boldsymbol{\pi}^k)$ denotes the updated policy when using $\boldsymbol{\pi}^k$ as the sampling policy. We express the trust region as that $D_{\mathrm{TV}}\left(\pi_i'(\cdot|s)||\pi_i^k(\cdot|s)\right) \leq \beta_i$. In this case, when $\beta_i \leq \frac{\sum_{s \in \mathcal{S}} \rho_{\boldsymbol{\mu}^*}(s)\phi(s, A_{\boldsymbol{\pi}^k})}{2 \sum_{s \in \mathcal{S}} \rho_{\boldsymbol{\pi}^k}(s)\phi(s, |A_{\boldsymbol{\pi}^k}|)}$, we have $J(\boldsymbol{\mu}^*, \boldsymbol{\mu}^*) \geq J(\boldsymbol{\pi}', \boldsymbol{\pi}^k)$.*

For proof see Appendix A.2. This theorem shows that when we replace $\boldsymbol{\mu}$ with an approximation of future teammate policies, under certain conditions we can at least obtain a smaller upper bound compared to the previous typical algorithms. Specifically, the required conditions are relevant to the trust-region setting.

## B  More Implementation Details

### B.1  More Details about Baselines

To conduct performance comparison in our experiments, we firstly select the main multi-agent actor-critic algorithms as baselines, including MADDPG (Lowe et al., 2017), IPPO (de Witt et al., 2020a), MAPPO (Yu et al., 2022a) and HAPPO (Kuba et al., 2021). Besides, we additionally design an opponent modeling algorithm TAPPO (abbreviated for Teammate-Aware MAPPO) to further compare our approach with traditional opponent modeling techniques. In specific, TAPPO learns teammate representations for extra policy conditions like in previous works (Papoudakis & Albrecht, 2020; Cao et al., 2023) and is incorporated into MAPPO. Furthermore, in the GRF environment, we additionally include the CDS algorithm (Li et al., 2021), which is a value-based algorithm specifically designed for the GRF problems. It mainly designs mechanism to enhance policy diversity among agents and surpasses typical value-based algorithms in the GRF environment in its experiments.

---

[3]$\psi(\boldsymbol{\mu}, \boldsymbol{\pi}^k)$ means the result of one round of policy update starting from $\boldsymbol{\pi}^k$ using $\boldsymbol{\mu}$ as the sampling policy, i.e., $\psi(\boldsymbol{\mu}, \boldsymbol{\pi}^k) = \arg\max_{\boldsymbol{\pi}} J(\boldsymbol{\pi}, \boldsymbol{\mu})$ within the trust region of $\boldsymbol{\pi}^k$ for MAPPO.

[4]Assuming $f$ is a function defined over the state and joint action space, the ego-max-operator $\phi$ is defined as $\phi(f, s) = \frac{1}{N} \sum_{i=1}^{N} \max_{a_i \in \mathcal{A}} \sum_{a_{-i} \in \mathcal{A}} \prod_{j \neq i} \pi_j^k(a_j|s) f(s, \boldsymbol{a})$.

## B.2 Incorporating Lookahead Strategy into HAPPO

Heterogeneous-Agent Proximal Policy Optimisation (HAPPO) (Kuba et al., 2021) is a recent work that introduced the *sequential policy update scheme* to the multi-agent policy gradient algorithm. It provides a monotonic improvement guarantee in theory based on the finding of the *multi-agent advantage decomposition lemma*. The core idea of this work is to update the agents' policies in sequence. This approach empowers subsequent agents to adapt their policies based on the updated strategies of preceding agents, thereby mitigating to some extent the impact of the "teammate delay" phenomenon. However, it has two main issues that might impact its effectiveness:

1) Despite the sequential policy update scheme, the preceding agents in the sequence still learn to cooperate with the previous round of teammates, which means the teammate delay issue persists for the preceding agents. This results in a critical importance placed on the order of agents' update (e.g., for the example in the Introduction section, if the order is Cook first, the updates are more efficient; otherwise, sequential update yields no benefits). However, HAPPO adopts random update orders, which poses a significant limitation.

2) Since in practice, the training trajectories are sampled by the policy of the previous round, HAPPO adopts Importance Sampling to help the subsequent agents learn to cooperate with updated previous agents. This approach can lead to a higher variance in the policy gradients as we need to multiply it by an importance sampling ratio to correct the objective. This issue becomes exacerbated when dealing with a larger number of agents, as we need to accumulate the product of importance ratios for all preceding agents.

Due to these two main issues, the effectiveness of HAPPO in practice may be influenced and it can not fully resolve the "teammate delay" issue . Actually, in practice, our lookahead strategy can be seamlessly integrated with HAPPO, further enhancing its effectiveness, which has been validated by our empirical experiments. The detailed process is introduced in the Algorithm 2, where the text highlighted in red emphasizes the uniqueness introduced by HAPPO.

---

**Algorithm 2** Heterogeneous-Agent Proximal Policy Optimisation with Lookahead

**Input**: The Number of agent $N$
**Output**: A cooperation policy for a multi-agent system

1: Initialize a replay buffer $\mathcal{B}$;
2: Initialize a policy $\boldsymbol{\pi}$ randomly;
3: **for** each iteration k **do**
4:     Sample a batch of transitions from $\mathcal{B}$ and update the environment model by minimizing loss $\mathcal{L}_{\mathrm{model}}$;;
5:     Sample a batch of trajectories $\tilde{\tau}$ in the environment model with sampling policy $\boldsymbol{\pi}^k$, and obtain $\tilde{\boldsymbol{\pi}}^{k+1}$ via maximizing $J(\boldsymbol{\pi}, \boldsymbol{\pi}^k)$ within trust region in a sequential update scheme using the training trajectories $\tilde{\tau}$;
6:     Sample a batch of trajectories $\tau$ in the real environment with sampling policy $\tilde{\boldsymbol{\pi}}^{k+1}$, and obtain $\boldsymbol{\pi}^{k+1}$ via maximizing $J(\boldsymbol{\pi}, \tilde{\boldsymbol{\pi}}^{k+1})$ within trust region in a sequential update scheme using the training trajectories $\tau$;
7:     Add trajectories $\tau$ to the buffer $\mathcal{B}$;
8: **end for**

---

## B.3 Details about Hyper-parameters

In this section, we firstly introduce the hyper-parameter configurations of our method in the experiments, and then we illustrate how we tune the hyper-parameters.

Table 5: Common hyper-parameters used across task scenarios of multi-agent MuJoCo. Note that lka is short for Lookahead, mlearn is short for "model learning", and ppo stands for Proximal Policy Optimization (PPO) algorithm.

| hyper-parameter | value | hyper-parameter | value | hyper-parameter | value |
|---|---|---|---|---|---|
| critic lr | 3e-4 | max grad norm | 10 | lka episode length | 20 |
| actor lr | 3e-4 | num rollouts | 40 | lka num mini-batches | 10 |
| gamma $\gamma$ | 0.99 | ppo num mini-batches | 10 | lka entropy coef | 0.001 |
| optimizer | Adam | entropy coef | 0.01 | mlearn batch size | 512 |
| optim eps | 1e-5 | stacked-frames | 1 | | |

### B.3.1 Hyper-parameter configuration

Here, we list the configuration of hyper-parameters that was utilized in our experiments to facilitate reproducing our experimental results. **Note** that for the common hyper-parameters, both the Lookahead algorithm and HAPPO adopted the same value in our experiments. Hence, the hyper-parameter configurations provided in this section are also applicable to our experimental results of the HAPPO algorithm.

**Multi-agent MuJoCo** For hyper-parameters that were set to the same values across all task scenarios, the configuration is provided in Table 5. Additionally, the varying hyper-parameter configurations across different tasks are provided in Table 6.

Table 6: Different hyper-parameters used across task scenarios of multi-agent MuJoCo. Note that lka is short for Lookahead, mlearn is short for "model learning", and ppo stands for Proximal Policy Optimization (PPO) algorithm.

| hyper-parameter | Ant | HalfCheetah | Walker2d |
|---|---|---|---|
| episode length | 200 | 400 | 200 |
| ppo num epochs | 10 | 20 | 10 |
| lka num rollouts | 2000 | 4000 | 2000 |
| lka num epochs | 10 | 20 | 10 |
| mlearn num epochs | 4000 | 2000 | 2000 |

**Google Research Football (GRF)** In the three task scenarios of Google Research Football (GRF), we employed identical hyper-parameter configurations, which are detailed in Table 7.

Table 7: Common hyper-parameters used across task scenarios of Google Research Football (GRF). Note that lka is short for Lookahead, mlearn is short for "model learning", and ppo stands for Proximal Policy Optimization (PPO) algorithm.

| hyper-parameter | value | hyper-parameter | value | hyper-parameter | value |
|---|---|---|---|---|---|
| critic lr | 5e-4 | episode length | 100 | lka num rollouts | 100 |
| actor lr | 5e-4 | num rollouts | 10 | lka num mini-batches | 1 |
| gamma $\gamma$ | 0.99 | ppo num mini-batches | 1 | lka num epochs | 15 |
| optimizer | Adam | ppo num epochs | 15 | lka entropy coef | 0 |
| optim eps | 1e-5 | entropy coef | 5e-3 | mlearn batch size | 1024 |
| max grad norm | 10 | lka episode length | 100 | mlearn num epochs | 800 |

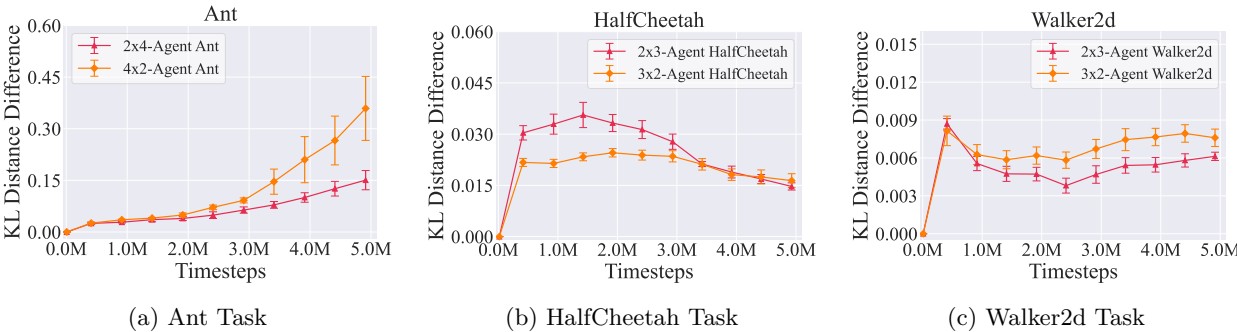

Figure 6: Measuring policy distances in multi-agent MuJoCo environment.

### B.3.2 Hyper-parameter tuning strategy

**Multi-agent MuJoCo** For the practical implementation efficiency of the algorithm, we employed JAX to implement our Lookahead algorithm. Additionally, to ensure a fair and effective comparison of the efficacy of our added lookahead strategy, the underlying HAPPO algorithm also utilized the same codebase. Furthermore, the fundamental hyper-parameters for both Lookahead and HAPPO were kept consistent. Consequently, we mainly tune the hyper-parameters to align the performance of the HAPPO algorithm of our codebase with that disclosed in the original paper.

**Google Research Football (GRF)** Similar to the case in multi-agent MuJoCo, to ensure a fair comparison, we maintained consistent foundational hyper-parameters for both Lookahead and the underlying HAPPO algorithm. While tuning these hyper-parameters, we conducted a search within certain ranges to fine-tune the underlying HAPPO algorithm for reasonably good performance results. Specifically, we explored learning rate `lr` (including `critic_lr` and `actor_lr`) within the range of {1e-4, 5e-4, 1e-3}, number of learning epochs `ppo_num_epochs` within {10, 15, 20}, and entropy regularization coefficient `entropy_coef` within {1e-3, 5e-3}.

## C More Experimental Results

### C.1 More Results about Lookahead Policy Analysis

In Section 4.3, we measure the KL distance difference of policies on Ant and HalfCheetah tasks of multi-agent MuJoCo. Here, we additionally provide the results on the Walker2d task. As we can see, the results on three different types of tasks all validate that our approach can empirically obtain positive results for $D_{\mathrm{KL}}(\pi^{k+1}, \pi^k) - D_{\mathrm{KL}}(\pi^{k+1}, \tilde{\pi}^{k+1})$, which means that $\tilde{\pi}^{k+1}$ can to some extent provide the direction information of future teammate policy $\pi^{k+1}$ and is relatively a good approximation.

### C.2 Study about Multiple Steps of Lookahead Approximation

In the practical implementation of the algorithm in this work, we employ a one-step approximation to estimate future teammate policies. We wonder whether we can obtain a better approximation of future teammates through more rounds of lookahead training. To answer this question, we conduct additional experiments, comparing with the baselines that perform more rounds of policy update when obtaining the lookahead policy. The results are depicted in Figure 7, where "Lookahead Step" represents the number of policy update rounds conducted for approximating future teammates, i.e., "Lookahead Step=1" corresponds to the results in the maintext.

From the results, we can see that when we increase the Lookahead Step, the quality of the lookahead approximation does not increase and we obtain lower performance. It is reasonable because when we conduct more iterations for lookahead training within the model, the newly updated policy would appear unfamiliar to the model, as the environmental model has been trained on data sampled from the old policies. This

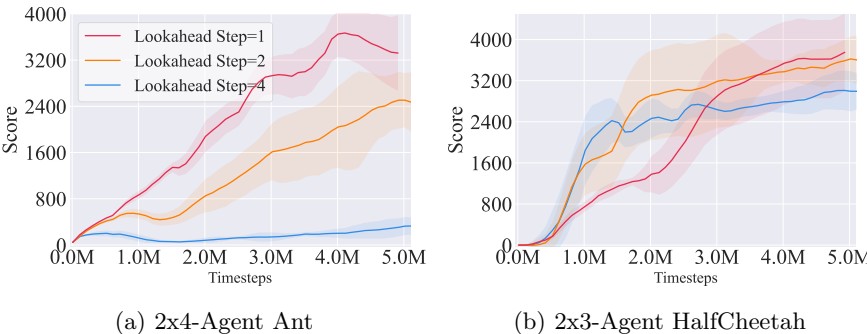

(a) 2x4-Agent Ant

(b) 2x3-Agent HalfCheetah

Figure 7: Experimental results for study about multiple steps of lookahead approximation.

necessitates refraining from employing excessively off-policy policies for trajectory rollout within the model. However, despite achieving lower convergence performance, it seems to learn faster in the early stage in the task of 2x3-Agent HalfCheetah when we increase the Lookahead Step. This encourages us to design better model learning algorithm in the future, thus to further improve the effectiveness of our approach.

