# OpenReview forum: "Efficient Multi-Agent Cooperation Learning through Teammate Lookahead"
_TMLR — Accepted by TMLR_

### Review · Reviewer_N2BW · 2024-12-07

**Summary Of Contributions:**

This paper investigates cooperative MARL. To improve learning stability and efficiency, this paper proposes an algorithm with a lookahead strategy, which enables the learning agents to predict the future behavior of teammates and make better policy updates. The authors provide theoretical results that justify the necessity of this lookahead strategy. The algorithm is tested in MuJoCo and Google Research Football experiments.

**Audience:**

Yes

**Broader Impact Concerns:**

Not applicable.

**Claims And Evidence:**

No

**Requested Changes:**

The aforementioned Weaknesses should be addressed. All of them are critical to securing my recommendation.

**Strengths And Weaknesses:**

Strength: The concept of predicting teammate policy updates is interesting. The provided algorithm is not computationally heavy, and can be seemlessly integrated with existing MARL frameworks.

Weaknesses on Theoretical Results:
- Theorem 1 doesn't sufficiently motivate us to solve for a sampling policy $\mu$ that minimizes term $(c)$. In Equation (7), with $\pi^{k+1}= \arg\max_{\pi}J(\pi,\mu)$ being a function of $\mu$, a natural target will be to find $\mu$ so that $J(\pi^{k+1}, \mu) - (c)$ is maximized. There is no reason to believe a $\mu$ that minimizes $(c)$ will maximize the two terms as a whole, unless the authors provide more intuitions.
- Theorem 2 is not rigirous. The definition of operator $\psi$ is unclear. In footnote (2), the ``trust region`` is ambiguous. The reviewer has checked the cited MAPPO paper, but it is not mentioned in the paper. Moreover, in the proof, the authors state that *$\psi$ typically means ... with some cinditions*. The terms ``typically`` and  ``some conditions`` are way too vague for a rigorous proof. Furthermore, the authors need to address the existence of such solution $\mu^*=\psi(\mu^*,\pi^k)$ in the proof.
- Aside from the aforementioned issues, the reviewer finds Theorem 2 trivial. Since $\pi^{k+1}$ is defined as $\psi(\mu, \pi^k)$ (footnote 2), an obvious $\mu$ closest to $\pi^{k+1}$ is simply the solution to $\mu = \psi(\mu,\pi^k)$, if such a solution exists. The reviewer suggests that the reviewer expressing the $\mu$ design in different way.

Other Weaknesses:
- Theorem 1 is not very informative. It appears to be a natural result that follows by Lemma 1 and a triangle inequality. It is better presented as a corollary.
- Some equations need to be checked. For example, In Equation (1), it seems there should be a factor $\frac{1}{1-\gamma}$ in the second term, if $\rho_{\pi}(s)$ is a probability distribution. In Equation (8), it seems $\min_{\mu}$ should be $\arg\min_{\mu}$.
- Some notations are confusing. The following are some examples.
    - At the beginning of Section 2.2, the reward function $\mathcal{R}$ is not explained. From the context, it seems the agents share the same reward function and this fact deserves a clarifying sentence.
    - $\eta(\pi)$ in Equation (1) is not formally defined throughout. According to the context, it should be the expect value $E[V_{\pi}(s)]$ over the initial state distribution. However, the initial state distribution is also not defined.
    - In Theorem 1, $\pi^*$ is not defined. We can not be sure whether it is Markovian or non-Markovian.

---

### Review · Reviewer_XofD · 2024-12-07

**Summary Of Contributions:**

This paper focuses on the drawback of existing cooperative MARL algorithms where agents update their individual policy without accounting for the concurrent updates in their teammates' policies. This phenomenon is termed as "teammate delay". In this work, a lookahead based approach is proposed that accounts for concurrent teammate updates by updating each agent's policy to learn to cooperate with predicted future teammates. It is a flexible algorithmic framework that can integrate with existing policy gradient based MARL algorithms, like HAPPO. Experiments with MA-MuJoCo and Google Research Football indicate that lookahead based policy update is competitive with prior MARL baselines and achieves similar evaluation performance with fewer training update steps.

**Audience:**

Yes

**Claims And Evidence:**

Yes

**Requested Changes:**

1. [Critical for decision] SMAC is another popular environment for testing MARL algorithms. Could the authors compare the proposed approach against baselines in SMAC?

2. [Critical for decision] I would request the authors to provide more details of their environment model - eg. the network architecture, training algorithm, etc.

3. [Critical for decision] Please fix the typos in the paper (eg. page 10: roll-outed -> rolled out; Abstract: learn to cooperative -> learn to cooperate).

4. [Critical for decision] The motivating example in Fig 1 and accompanying text in the first paragraph of page 2 is confusing to me. This particular example changes the cook's task or goal (from cooking hamburgers to cooking salad) which causes the mismatch between the teammates policies. This, to me, does not seem like a convincing example of "teammate delay" because the rest of the paper focuses on settings where all teammates are updating their policy towards the same goal.

**Strengths And Weaknesses:**

**Strengths**

This paper motivates a practical lookahead based approach to policy learning in MARL through theoretical intuitions.

1. _Regret analysis to quantify the impact of teammate delay_: Lemma 1 computes an upper bound for the difference between the surrogate loss $J(\pi^{k+1},\mu)$ used during training update (at $\pi^k$) and the actual return $\eta(\pi^{k+1})$ evaluated in practice. The bound depends on the total variation distance between the sampling $(\mu)$ and updated $(\pi^{k+1})$ policies. Theorem 1 establishes an upper bound for the overall regret of the updated joint policy $\pi^{k+1}$, showing that this bounding term can be minimized by reducing the mismatch between $\mu$ and $\pi^{k+1}$.

2. _Future teammate approximation through one-step lookahead_: Due to the coupling between $\mu$ and $\pi^{k+1}$, the authors propose a bilevel optimization problem that solves for the sampling policy $\mu^*$ which would minimize the TV distance between $\mu$ and $\pi^{k+1}$ in the previous regret analysis. Following similar approaches to bi-level leader follower problems with Stackelberg games, this paper proposes an optimization step that first solves for a sampling policy $\tilde{\pi}^{k+1}$ which approximates future teammate policy and uses $\tilde{\pi}^{k+1}$ to estimate $\pi^{k+1} = \mathop{argmax}_{\pi} J(\pi, \tilde{\pi}^{k+1})$.

3. _Leveraging model based RL to improve training sample efficiency_: To estimate future teammate policies efficiently, the lookahead approximation method is integrated into a model-based learning framework. Trajectories collected by the sampling policy during policy update are stored in a replay buffer and used to train the environment model.


4. _Ablation studies for analyzing benefits of lookahead over baselines_: Comparisons with baseline methods like MADDPG, MAPPO and HAPPO validate the effectiveness of the lookahead strategy. The proposed method easily integrates with existing algorithms like MAPPO and HAPPO without requiring substantial architectural changes, making it accessible for adoption.


**Weaknesses**

1. _Performance limited by model error_: As the authors also admit, the "practical algorithm performance is to some extent limited by the model learning error". Assuming the authors have used a predictive neural network model of the environment (note request for architecture and training details under Requested Changes), it would have to be retrained to model different environments. The authors have not discussed the potential of using their proposed approach with methods from generative modeling of the environment (eg. maybe drawing from the ongoing research in generative world models).

2. _Scalability with larger teamsizes_: The experiments are limited to settings with almost 3 agents. The current set of results do not provide sufficient evidence of performance in environments with larger teamsizes where approximating the future teammate policy would be challenging. Particularly, there is a lack of discussion of this method's limitations with stochastic policies.

3. _Applicability to POMDPs_: The ability to leverage a well trained environment model would be particularly important when the environment is partially observable. In this paper, authors focus primarily on MDP environments and propose model learning methods for future work.

4. _Performance in long horizon tasks_: This paper does not focus on tasks that require multi-agent cooperation over long horizons and/or against strategic opponents, so it is difficult to argue for the practical utility of the proposed method in realistic application settings.

---

### Review · Reviewer_7aYm · 2024-12-18

**Summary Of Contributions:**

This paper addresses the “teammate delay” issue in Cooperative Multi-Agent Reinforcement Learning (MARL), where agents experience discrepancies due to concurrent teammate updates. By introducing a lookahead strategy to predict future teammates, the approach improves learning stability and efficiency, achieving superior performance over state-of-the-art MARL methods.

**Audience:**

Yes

**Broader Impact Concerns:**

Nan

**Claims And Evidence:**

Yes

**Requested Changes:**

Please see my comments on weaknesses.

**Strengths And Weaknesses:**

The method used to deal with the “teammate delay” issue is very interesting. However, I find several concerns regarding the problem's validity, theoretical clarity, and overall presentation. My detailed comments are as follows:

The “teammate delay” described in the paper seems to be a potentially artificial issue that arises due to specific assumptions about the learning setup. Its significance is questionable for the following reasons:

The multi-agent environment is inherently dynamic because each agent’s strategy influences others. This non-stationarity arises naturally from agents’ concurrent updates rather than a specific "delay" mechanism.
1.If agents update their strategies simultaneously, there is no actual "lag." Agents will adapt to updated teammate policies during interaction. In distributed or asynchronous optimization, delays are more related to computation or data synchronization, not learning.
2.In cooperative MARL, agents dynamically adjust to others’ behaviors through feedback mechanisms (e.g., rewards or shared information). While “teammate delay” might temporarily influence progress, agents’ strategies are expected to converge over time.
3.The paper builds on Equation (4) without introducing the Performance Discrepancy Lemma in the preliminaries. Readers unfamiliar with this result might find the subsequent theoretical analysis difficult to follow. More background on this lemma should be provided.

The definition of  (state distribution under the discounted return) is missing from the main text and is instead deferred to the appendix, increasing the reading difficulty. However, even in the appendix, there is no explicit mathematical definition of it.

While the result in Equation (2) can be supported by prior literature, the authors fail to rigorously justify Equation (3). The derivation of Equation (3) is not presented, and the paper does not provide a strict proof. Please include a detailed derivation to validate this result.

The rationale behind Equation (5) is insufficiently explained. The text vaguely states it is for “further analysis,” but this does not justify its introduction. Moreover, introducing time indices could improve the clarity of this equation.

The significance of the bound presented in Lemma 1 is not well explained. Please elaborate on the practical implications of this result.

The paper refers to a policy update operator and an ego-max operator, but these terms are not formally defined in the main text. Definitions should be provided for clarity.

The proof of Theorem 2 is written ambiguously. Specifically, the statement “ \pi^k+1 can be defined” lacks justification, and the use of subscript k is inconsistent throughout the proof. This inconsistency makes the proof difficult to follow and undermines its rigor. Additionally, the explanation for why \pi^k+1 = \mu^* holds is not clear. This part of the proof requires more formal reasoning.

---

### Decision · Action_Editor_yReE · 2025-02-05

**Recommendation:** Accept with minor revision

**Comment:**

This paper addresses the “teammate delay” issue in Cooperative Multi-Agent Reinforcement Learning (MARL), where agents experience discrepancies due to concurrent teammate updates. By introducing a lookahead strategy to predict future teammates, the approach improves learning stability and efficiency, achieving superior performance over state-of-the-art MARL methods.

All the reviewers acknowledge the contribution of the work, while there are several minors raised by reviewers should be addressed in final version:

- Some claims in the paper is over-claiming, "The empirical results unequivocally demonstrate the superiority of our method,..", however, there exists competitors on par with the proposed algorithm.

- The terminology "teammate delay problem" is related to the instability issues of multi-agent training environments, which should be discussed to avoid potential confusion.

**Audience:**

Yes. The multi-agent reinforcement learning community will be interested in this work.

**Claims And Evidence:**

The high-level idea is intuitive and easy to implement, and the authors demonstrated the performances empirically to justify the claim.